# S³NET: STAGE-AWARE SLEEP STAGING NETWORK

## ABSTRACT

Automated sleep staging is a critical component in the diagnosis of sleep disorders and the analysis of sleep architecture. While deep learning approaches that leverage time-frequency representations have shown promise, their performance remains suboptimal, primarily due to two fundamental limitations: (1) the inability to effectively model the subtle distinctions of transitional sleep stages (N1 and N2), which exhibit ambiguous electrophysiological patterns, and (2) the inefficient fusion of complementary information from time-domain and frequency-domain representations. To this end, we propose S³Net, a novel Stage-Aware Sleep Staging Network that introduces two dedicated components to address these challenges. First, a Stage-Aware Experts (SAE) module explicitly partitions the sleep stages into easy- and hard-to-separate groups, processing them through separate expert network branches. This allows for specialized feature refinement, particularly for the challenging transitional stages. Second, to foster a cohesive representation, we design a Time Alignment Module (t-ALN) that projects frequency-derived features onto the time axis, effectively bridging the domain gap and enabling synergistic integration of multi-view features. We evaluate S³Net on three public polysomnography datasets (ISRUC-S1, ISRUC-S3, and Sleep-EDF-153). Our model consistently sets a new state-of-the-art, achieving an overall accuracy of 85.6%, 86.6%, and 86.9%, respectively, and demonstrates a noticeable improvement in classifying the N1 and N2 stages. The results validate the efficacy of our stage-aware design and structural alignment strategy, offering a more robust framework for clinical and portable sleep staging. Source code is available at `https://anonymous.4open.science/r/S3Net/`.

## 1 INTRODUCTION

Accurate sleep stage classification is a critical diagnostic tool in clinical neurophysiology and a fundamental task in sleep research (Sheybani et al., 2025). The prevailing methodology relies on polysomnography (PSG), which captures multi-modal electrophysiological time series—notably electroencephalography (EEG), electrocardiogram (ECG), electrooculography (EOG), and electromyography (EMG) signals. These data are manually annotated by sleep technologists into the five stages (W, N1, N2, N3, REM) defined by the American Academy of Sleep Medicine (AASM) standard (Berry et al., 2017). This manual scoring paradigm, however, introduces significant bottlenecks that impede scalability and objectivity. The process is inherently labor-intensive and exhibits considerable inter-rater variability, even among experts (Yang et al., 2025). More fundamentally, manual analysis is ill-suited to modeling the complex, non-linear temporal dynamics and high-dimensional interactions present in raw PSG data. These limitations motivate the development of automated staging systems capable of leveraging the full information of the signal for objective sleep analysis.

The demand of automated sleep staging has catalyzed the development of diverse deep learning paradigms, which can be broadly categorized by their approach to modeling temporal and spectral information. Initial efforts focused on **temporal feature extractors**, primarily employing Convolutional and Recurrent Neural Networks (CNNs and RNNs) (Supratak et al., 2017; Supratak & Guo, 2020; Jia et al., 2020b; Phan et al., 2019; 2022). These models are effective at identifying localized characteristic waveforms (e.g., spindles, K-complexes) and learning short-to-medium-range contextual transitions between stages. While offering a practical trade-off between model capacity and computational cost, their ability to capture long-range dependencies is inherently limited by sequential processing or finite receptive fields.

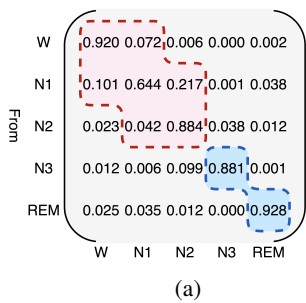
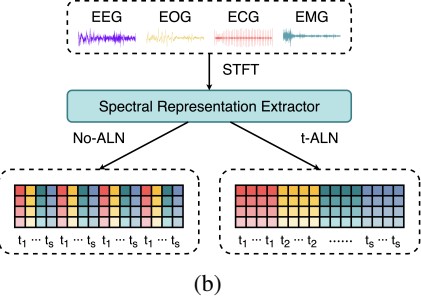

|     | (a) |     |     | (b) |

Figure 1: (a) Sleep stage transition matrix, and (b) Time position of spectral representation of ResNet. Left branch (No-ALN) denotes the feature maps without alignment while right branch (t-ALN) with time alignment.

To address this constraint, a second paradigm of **long-range context modelers** based on Transformer architectures has gained prominence (Ji et al., 2024; Zhou et al., 2025; Liu & Jia, 2023). By leveraging self-attention mechanisms, these models can integrate information across the entire recording, enabling globally coherent stage predictions and a more natural fusion of multi-channel inputs. However, the self-attention mechanism is computationally intensive and operates on a pre-defined feature space, which may underutilize the well-established spectral characteristics of sleep.

This limitation has motivated a third, distinct approach: **spectral-domain analysis**. Here, signals are transformed into time-frequency representations (e.g., via spectrograms or wavelets) before being processed, often by CNN-based architectures (Peng et al., 2023; Liang et al., 2025; Fei et al., 2024; Yang et al., 2025; Li et al., 2022). This transformation explicitly encodes clinically fundamental oscillatory rhythms (e.g., alpha, sigma, delta power) that are convolutional priors for sleep staging, making them more salient than in the raw time domain. Furthermore, it provides a unified, physiologically meaningful space for aligning and fusing heterogeneous signal modalities.

Despite considerable progress, automated sleep staging models remain constrained by two fundamental limitations. First, current architectures exhibit ineffective modeling of subtle inter-stage transitions, particularly between light sleep stages N1 and N2. The transition matrix in Figure 1a shows that N3 and REM form a relatively stable regime, as reflected by high self-transition probabilities and minimal cross-transitions. In contrast, N1 exhibits pronounced instability, with elevated flows toward adjacent stages W and N2, highlighting its role as a transitional state between wakefulness and light sleep. This performance disparity stems from a critical architectural mismatch: conventional approaches employ a monolithic processing pipeline that fails to account for the heterogeneous discriminative complexity across sleep stages. These models allocate uniform representational capacity to all stages, resulting in insufficient modeling power for the subtle, low-signal-to-noise-ratio patterns characterizing transitional stages (N1/N2), while over-parameterizing the classification of highly distinctive waveforms in stages like N3 and REM.

Second, existing methods show ineffective cross-domain integration of time-frequency representations. As shown in Figure 1b, contemporary approaches typically employ CNNs to learn frequency representations and Transformers to model temporal dependencies, while they suffer from a fundamental representational misalignment. Frequency representations extracted by CNNs are typically flattened before being fed to Transformer networks, which disrupts their inherent temporal structure and positional coherence. Since Transformer networks critically rely on precise positional embeddings for modeling temporal relationships, this flattening operation causes a disintegration of temporal alignment between frequency-domain and time-domain representations. Consequently, the model's capacity for effective cross-domain feature interaction is severely compromised, leading to suboptimal fusion of complementary information and ultimately degrading staging performance.

To address these challenges, we introduce $S^3$Net, a novel sleep staging framework that integrates two core innovations designed to overcome fundamental limitations in existing approaches. First, our Time Alignment Module (t-ALN) explicitly resolves the temporal representation mismatch between spectral and temporal domains by projecting frequency-derived features onto the temporal axis through a learnable alignment operation, enabling coherent cross-domain fusion. Second, our Stage-Aware Experts (SAE) module dynamically modulates representational weights across two specialized branches to address the heterogeneous complexity of sleep stages. The hard-separated

expert is tailored to capture the subtle and transitional dynamics of stages W, N1, and N2, while the easy-separated expert is designed to characterize the distinctive and more separable patterns of N3 and REM. We validate $S^3$Net through extensive experiments on three public benchmarks (ISRUC-S1, ISRUC-S3, and Sleep-EDF-153), demonstrating state-of-the-art performance, with a noticeable improvement in the classification of transitional stages. Furthermore, our method provides both quantitative improvements and qualitative insights into sleep stage dynamics. To facilitate reproducibility, all code and preprocessed data will be publicly released. Overall, the key contributions of our work are summarized as follows:

- We propose t-ALN, a novel temporal alignment module that bridges the spectral-temporal representation mismatch in sleep staging through a learnable projection. Furthermore, t-ALN as an information bottleneck constrains the temporal representation, thereby implicitly preserving features relevant to sleep stage classification.

- We introduce SAE, a stage-aware experts module, which is a dynamic architecture that employs two specialized experts to address the heterogeneous complexity of sleep staging. One expert captures the subtle dynamics of transitional stages (W, N1, N2), while the other distinguishes the distinct stages of N3 and REM, leading to more robust and discriminative feature learning.

- Empirical results demonstrate that $S^3$Net sets a new state-of-the-art on three three major sleep staging datasets of ISRUC-S1, ISRUC-S3, and Sleep-EDF-153. Furthermore, $S^3$Net achieves a steady reduction in inference time, as experimentally validated.

- The design choices of $S^3$Net are rigorously validated by comprehensive ablation studies, while the model's interpretability is substantiated through visualizations such as alignment maps, and analyses of expert routing behavior.

## 2 RELATED WORK

### 2.1 SLEEP STAGE CLASSIFICATION

Sleep stage classification is a long-standing task in biomedical signal processing, where deep models have shown strong potential in learning from complex, non-stationary PSG signals. Early CNN- and RNN-based methods (Chen et al., 2023a; Jia et al., 2020a; Lee et al., 2024; Shen et al., 2024; Phan et al., 2019; 2022) effectively captured local patterns and mid-range events like spindles and K-complexes, but struggled with long-range dependencies due to limited receptive fields. Transformer-based models, such as MixSleepNet (Ji et al., 2024), trans-SF-UIDA (Zhou et al., 2025), and BSTT (Liu & Jia, 2023), addressed this by modeling broader temporal context via self-attention. However, many still underutilize frequency-domain cues—like slow waves in N3 or REM's fast oscillations—which are vital for accurate classification. To better integrate temporal and spectral views, recent methods like MVF-SleepNet (Li et al., 2022), cVAN (Yang et al., 2025), and SPTESleepNet (Chen et al., 2024) introduced multi-view fusion or spectral embeddings, while WASR (Fei et al., 2024) reconstructed frequency features dynamically. Yet, most rely on early or implicit fusion and lack structural alignment across domains. In contrast, we propose a time Alignment module called t-ALN for projecting spectral features onto the temporal axis, and a Cross Swin Transformer (CST) to model cross-view dependencies for more robust and fine-grained classification.

### 2.2 MIXTURE-OF-EXPERTS

Mixture-of-Experts (MoE) has been widely used in computer vision (Yu et al., 2024; Chen et al., 2023b), time-series forecasting (Shi et al., 2025), and natural language processing (Zhao et al., 2024), where routing inputs to specialized experts enables both model scaling and performance gains. This mechanism also improves interpretability by encouraging different experts to focus on distinct input patterns. Despite its effectiveness, MoE remains underexplored in physiological signal analysis. A recent exception is Seizure-MoE (Du et al., 2023), which applies MoE to epileptic seizure detection and shows the advantage of expert specialization in modeling pathological EEG activity. However, in sleep stage classification, MoE has been rarely used. Given the structured yet asymmetric nature of sleep transitions, we draw inspiration from the expert specialization paradigm

in MoE and propose Stage-Aware Experts, which construct highly task-driven branches dedicated to capturing stage-specific dynamics. By allocating sleep stages to two specialized expert pathways and combining their outputs through Cross-Gate derived soft weights, the framework dynamically models both stage-specific patterns and transition variability.

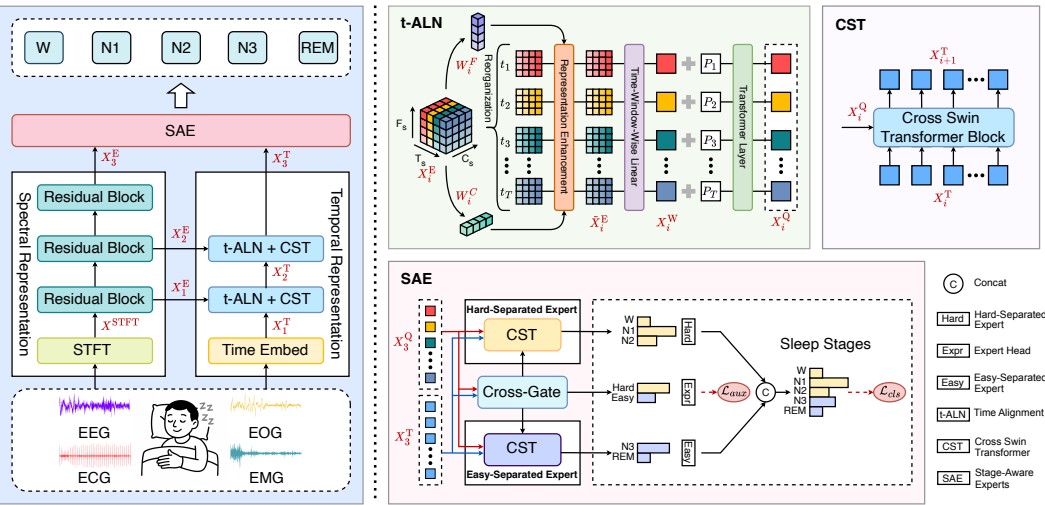

Figure 2: An overview of the S$^3$Net architecture. It consists of a temporal representation extractor, a spectral representation extractor, and a stage-aware experts (SAE).

## 3 METHODOLOGY

### 3.1 PRELIMINARIES

The task of sleep stage classification typically relies on overnight PSG recordings, which we denote as a dataset $\mathcal{S} = \{(X_i, Y_i)\}_{i=1}^N$, where $X_i$ represents an overnight PSG signal and $Y_i$ represents the corresponding label sequence of the $i$-th subject. Each $X_i$ is segmented into $L$ consecutive 30-second segments, i.e., $X_i = \{X^l\}_{l=1}^L$, where each segment $X^l = \{\mathbf{x}^1, \ldots, \mathbf{x}^C\}$ is a multivariate time series across $C$ channels and $\mathbf{x}^c \in \mathbb{R}^T$ denotes the signal from the $c$-th channel. The label sequence is $Y_i = \{y^l\}_{l=1}^L$, with $y^l \in \{0, 1, 2, 3, 4\}$ indicating one of five standard stages: W, N1, N2, N3, and REM. Additionally, we define an auxiliary label $y_{\text{expert}}^l \in \{0, 1\}$ for each segment, where 0 corresponds to hard-separated stages (W, N1, N2) and 1 to easy-separated stages (N3, REM), which serves to supervise the stage-aware expert module.

### 3.2 OVERVIEW OF S$^3$NET

S$^3$Net is a sophisticated deep learning framework designed for automated sleep stage classification, engineered to overcome the fundamental challenges of fusing multi-modal physiological data and accurately discriminating between sleep stages of varying complexity. As illustrated in the Figure 2, the architecture follows a logical, multi-stage pipeline that progresses from raw signal input to the final stage prediction, integrating several innovative components along the way.

The process begins with the simultaneous processing of multiple input signals, such as EEG, EOG, ECG and EMG. These signals are processed through two parallel, dedicated pathways to extract complementary representations. In the spectral pathway, the raw segment $X^l$ is transformed into the time-frequency domain using a Short-Time Fourier Transform (STFT) to yield $X^{\text{STFT}}$, capturing salient rhythmic patterns. Concurrently, the temporal pathway processes the original time-domain segment $X^l$, through convolutional embedding layers, to produce temporal feature $X_1^T$, preserving the sequential dynamics and morphological features essential for understanding sleep architecture.

Based on the spectral representation extractor, the input $X^{\text{STFT}}$ is transformed into frequency-domain features through three Residual Blocks composed of convolutional layers, which progres-

sively capture multi-level time–frequency representations. To enhance interpretability and guide the model's focus toward physiologically meaningful patterns, the extracted features are element-wise squared to yield spectral energy representations, denoted as $[X_1^E, X_2^E, X_3^E]$. These energy representations encode localized energy distributions across both time and frequency dimensions.

The temporal representation extractor is constructed upon the Cross Swin Transformer (CST, see Appendix A.3), inspired by the original Swin Transformer (Liu et al., 2021) design but adapted to enable effective cross-domain fusion. This module integrates temporal and spectral features while simultaneously capturing localized short-term dynamics (e.g., spindles) and long-range dependencies spanning entire sleep cycles. To further alleviate misalignment between temporal and spectral characteristics, a dedicated Time Alignment operation is introduced, generating temporally aligned spectral representations $[X_1^Q, X_2^Q, X_3^Q]$ from the corresponding energy features $[X_1^E, X_2^E, X_3^E]$. These aligned representations preserve sequential coherence and provide a structured basis for cross-domain interaction. Within this formulation, $X_1^Q$ is employed as the query, while the temporal embedding $X_1^T$ serves as both key and value. Leveraging the expressive modeling capacity of the Cross Swin Transformer, the framework progressively integrates information by fusing $X_1^Q$ with $X_1^T$ to obtain $X_2^T$, and iteratively extending this process to yield $X_3^T$, resulting in refined temporal representations that unify information from both time and frequency domains.

Once both temporal and spectral representations have been derived, Stage-Aware Experts (SAE) are introduced to explicitly disentangle the heterogeneous discriminative difficulty across sleep stages. A hard-separated expert is dedicated to capturing the subtle, low-signal-to-noise-ratio patterns and highly transitional characteristics that define transitional stages (W, N1, N2), with particular emphasis on the instability of N1, which is especially prone to transitions into other stages. In contrast, an easy-separated expert is responsible for distinguishing stages with highly distinctive waveforms, such as N3 and REM, thereby preventing the excessive allocation of capacity to relatively separable states. This expert specialization enables adaptive distribution of representational resources according to stage-specific complexity, relying on the refined temporal representation $X_3^T$ together with the final aligned spectral representation $X_3^Q$. To further consolidate decision reliability, a Cross-Gate mechanism is incorporated to softly weight the outputs of the two experts, facilitating input-adaptive fusion and producing the final prediction.

### 3.3 TIME ALIGNMENT

As illustrated in the t-ALN component of Figure 2, the t-ALN connects the spectral branch to the temporal branch. It reduces the mismatch between time- and frequency-domain features by mapping spectral energy maps into a compact, temporally organized query representation. This mapping is implemented as a learnable projection consisting of a Time-Window-Wise Linear layer and a Transformer layer. Specially, to selectively emphasize informative structures in the energy maps $X_i^E \in \mathbb{R}^{C_s \times F_s \times T_s}, i \in \{1, 2, 3\}$, we aggregate activations along the frequency and channel dimensions to obtain frequency-wise and channel-wise energy statistics, which are then normalized with a softmax function to produce two sets of attention weights: a frequency-wise weight $W_i^F \in \mathbb{R}^F$ and a channel-wise weight $W_i^C \in \mathbb{R}^C$, defined as

$$W_i^F = \text{softmax}\left(\frac{1}{C_s T_s} \sum_{c=1}^{C_s} \sum_{t=1}^{T_s} X_i^E\right), \quad W_i^C = \text{softmax}\left(\frac{1}{F_s T_s} \sum_{f=1}^{F_s} \sum_{t=1}^{T_s} X_i^E\right). \quad (1)$$

These weights are then applied to enhance the spectral energy along their respective dimensions:

$$\tilde{X}_i^E = X_i^E \cdot W_i^F \cdot W_i^C, \quad (2)$$

producing an energy-refined representation $\tilde{X}_i^E \in \mathbb{R}^{T_s \times F_s \times C_s}$ that highlights frequency- and channel-specific patterns. Following this refinement, the enhanced energy maps are passed through a Time-Window-Wise Linear layer, which performs a linear projection on the channel dimension within each time window, yielding a more compact channel representation. To preserve the temporal identity of each time step before flattening, stepwise positional encodings are incorporated along the time axis so that time-specific information remains distinguishable in the subsequent flattened representation. The resulting projected features $X_i^W \in \mathbb{R}^{T_s \times F_s \times D_s}$ thus encode time-specific information in a form that supports structured interaction across time and frequency domains. The

projected features are then flattened by concatenating the representations of successive time steps,

$$\tilde{X}_i^W = \left[ \, X_i^W(1,:,:), \; X_i^W(2,:,:), \; \ldots, \; X_i^W(T_s,:,:) \, \right]. \qquad (3)$$

so that all features belonging to the same time step are associated with an identical temporal positional embedding after flattening. This sequence $\tilde{X}_i^W \in \mathbb{R}^{(T_s \cdot D_s) \times F_s}$ is subsequently processed by a Transformer layer to model contextual dependencies across frequency bands. The Transformer outputs $X_i^Q$, a temporally aligned and spectrally refined query representation that conforms to the input format required by the CST and serves as the final output of t-ALN for downstream sleep stage classification. When $X_i^Q$ is used as the query in cross-attention with the temporal features, it further acts as an information bottleneck on the temporal pathway: only temporal patterns that are consistent with the spectro-temporal cues encoded in $X_i^Q$ receive high attention weights, which suppresses noise from temporal representation and preserves discriminative features for sleep staging. The t-ALN algorithm and its theoretical analysis are described in Appendices A.2 and A.18, respectively.

### 3.4 STAGE-AWARE EXPERTS

To accommodate the complex transition dynamics of sleep stages—particularly the hard-separated N1 stage—we introduce the Stage-Aware Experts (SAE) module, as illustrated in figure 2. Given the varying degrees of separability across stages, we design two specialized expert branches: a Hard-Separated Expert for transitional stages (W, N1, N2) and a Easy-Separated Expert for stable stages (N3, REM). Each expert comprises a CST block (without patch merging) and a classifier head. The temporal representation $X_3^T$ and the temporally aligned spectral representation $X_3^Q$ are jointly fed into both experts:

$$\begin{aligned} y_{\text{Hard}} &= \text{Hard-Separated Expert}(Q = X_i^Q, \; K = X_i^T, \; V = X_i^T), \\ y_{\text{Easy}} &= \text{Easy-Separated Expert}(Q = X_i^Q, \; K = X_i^T, \; V = X_i^T). \end{aligned} \qquad (4)$$

To dynamically allocate their outputs, we employ a Cross-Gate mechanism, where $X_3^Q$ serves as the query and $X_3^T$ as the key and value:

$$\begin{aligned} \hat{y}_{\text{Expert}}^l, \; w_{\text{Hard}}, \; w_{\text{Easy}} &= \text{Cross-Gate}(Q = X_3^Q, \; K = X_3^T, \; V = X_3^T), \\ \hat{y}^l &= \text{Concat}(w_{\text{Hard}} \cdot y_{\text{Hard}}, \; w_{\text{Easy}} \cdot y_{\text{Easy}}). \end{aligned} \qquad (5)$$

Here, $\hat{y}_{\text{expert}}^l$ denotes the gating logits produced by the Cross-Gate module. These logits are passed through a softmax layer to derive the expert weights $w_{\text{Hard}}$ and $w_{\text{Easy}}$. The final five-class prediction $\hat{y}^l$ is formed by weighted concatenation of the expert outputs. To jointly supervise both the final prediction and the expert routing behavior, we adopt a hybrid loss comprising a five-class classification loss for the final prediction and a binary classification loss for the expert routing:

$$L_{\text{total}} = L_{cls} + \alpha L_{aux} = \text{CE}(\hat{y}^l, y^l) + \alpha \text{CE}(\hat{y}_{\text{Expert}}^l, y_{\text{Expert}}^l). \qquad (6)$$

where $L_{aux}$ encourages the Cross-Gate module to assign inputs to the appropriate expert. A balancing coefficient $\alpha$ is used to combine the two terms. In addition, Appendix A.17 provides a theoretical analysis demonstrating the effectiveness of SAE, with supporting empirical validation.

## 4 EXPERIMENTS

### 4.1 IMPLEMENT DETAILS

All experiments are conducted using Python 3.11 and PyTorch 2.6.0. The training and evaluation processes are performed on a workstation with four NVIDIA RTX A6000 GPUs, with 10,752 CUDA cores and 48GB of VRAM. The system is powered by an Intel Xeon Platinum 8474C CPU with 512GB of system memory. Architectural and training details are provided in Appendix A.5.

### 4.2 DATASETS

We evaluate S$^3$Net on three publicly available overnight PSG datasets: **ISRUC-S1**, **ISRUC-S3**, and **Sleep-EDF-153**. All recordings are segmented into 30-second segments and annotated according to

AASM or R&K standards. For the ISRUC datasets (S1 and S3), we use the same set of 10 channels comprising EEG, EOG, EMG, and ECG signals, while for Sleep-EDF-153 we use two EEG channels as inputs. In total, we use 87,187, 8,589, and 195,292 sleep segments from ISRUC-S1, ISRUC-S3, and Sleep-EDF-153, respectively. Further dataset details are provided in the Appendix A.7.

## 4.3 BASELINES

We compare $S^3$Net with 15 representative sleep staging models, covering various temporal architectures. These include DeepSleepNet (Supratak et al., 2017), TinySleepNet (Supratak & Guo, 2020), XSleepNet (Phan et al., 2022), SleePyCo (Lee et al., 2024), SeqSleepNet (Phan et al., 2019), MVF-SleepNet (Li et al., 2022), DGraphormer-SleepNet(Huang et al., 2025), cVAN (Yang et al., 2025), STGCN (Jia et al., 2020b), MSTGCN (Jia et al., 2020a), StAGN (Chen et al., 2023a), MixSleep-Net (Ji et al., 2024), BSTT (Liu & Jia, 2023), SLEEPSMC (Ma et al., 2025), and CIMSleepNet (Shen et al., 2024). Full details of the baselines are provided in the Appendix A.8.

Table 1: Performance comparison with state-of-the-art sleep staging methods across datasets.

| Dataset | Model | Acc | F1 | $\kappa$ | F1 score for per stage | | | | |
|---------|-------|-----|-----|----------|------|------|------|------|------|
| | | | | | W | N1 | N2 | N3 | REM |
| ISRUC-S1 | SLEEPSMC(Ma et al., 2025) | 0.771 | 0.746 | 0.702 | 0.886 | 0.480 | 0.751 | 0.814 | 0.800 |
| | MSTGCN(Jia et al., 2020a) | 0.809 | 0.787 | 0.752 | 0.893 | 0.531 | 0.799 | 0.867 | 0.844 |
| | StAGN(Chen et al., 2023a) | 0.811 | 0.790 | - | 0.895 | 0.547 | 0.797 | 0.876 | 0.836 |
| | DGraphormer-SleepNet(Huang et al., 2025) | 0.814 | 0.788 | - | 0.907 | 0.511 | 0.800 | 0.874 | 0.846 |
| | BSTT(Liu & Jia, 2023) | 0.820 | 0.803 | 0.768 | - | - | - | - | - |
| | MVF-SleepNet(Li et al., 2022) | 0.821 | 0.802 | 0.768 | 0.908 | 0.562 | 0.811 | 0.871 | 0.857 |
| | MixSleepNet(Ji et al., 2024) | 0.829 | 0.791 | 0.755 | 0.903 | 0.482 | 0.826 | 0.878 | 0.868 |
| | cVAN(Yang et al., 2025) | 0.835 | 0.821 | 0.788 | 0.914 | 0.599 | 0.826 | 0.896 | 0.872 |
| | **S³Net** | **0.856** | **0.842** | **0.814** | **0.933** | **0.628** | **0.840** | **0.910** | **0.900** |
| ISRUC-S3 | SeqSleepNet(Phan et al., 2019) | 0.789 | 0.763 | 0.725 | 0.836 | 0.439 | 0.793 | 0.879 | 0.867 |
| | STGCN(Jia et al., 2020b) | 0.799 | 0.787 | 0.741 | 0.878 | 0.574 | 0.776 | 0.864 | 0.841 |
| | SLEEPSMC(Ma et al., 2025) | 0.793 | 0.782 | 0.734 | 0.876 | 0.572 | 0.775 | 0.872 | 0.812 |
| | MSTGCN(Jia et al., 2020a) | 0.821 | 0.808 | 0.769 | 0.894 | 0.596 | 0.806 | 0.890 | 0.856 |
| | MVF-SleepNet(Li et al., 2022) | 0.841 | 0.828 | 0.795 | 0.900 | 0.625 | 0.833 | 0.911 | 0.873 |
| | StAGN(Chen et al., 2023a) | 0.844 | 0.836 | - | 0.907 | 0.663 | 0.832 | 0.895 | 0.881 |
| | DGraphormer-SleepNet(Huang et al., 2025) | 0.854 | 0.845 | - | 0.919 | 0.676 | 0.838 | 0.907 | 0.883 |
| | cVAN(Yang et al., 2025) | 0.856 | 0.842 | 0.810 | 0.915 | 0.674 | 0.844 | 0.911 | 0.864 |
| | **S³Net** | **0.866** | **0.855** | **0.827** | **0.924** | **0.678** | **0.858** | **0.927** | **0.887** |
| Sleep-EDF-153 | DeepSleepNet(Supratak et al., 2017) | 0.785 | 0.753 | - | 0.910 | 0.470 | 0.810 | 0.690 | 0.790 |
| | SLEEPSMC(Ma et al., 2025) | 0.816 | 0.756 | 0.745 | 0.924 | 0.429 | 0.836 | 0.800 | 0.791 |
| | TinySleepNet(Supratak & Guo, 2020) | 0.831 | 0.781 | - | 0.928 | 0.510 | 0.853 | 0.811 | 0.803 |
| | SeqSleepNet(Phan et al., 2019) | 0.838 | 0.789 | - | 0.929 | 0.489 | 0.854 | 0.786 | 0.851 |
| | XSleepNet(Phan et al., 2022) | 0.840 | 0.779 | 0.778 | - | - | - | - | - |
| | SleePyCo(Lee et al., 2024) | 0.846 | 0.790 | 0.787 | 0.935 | 0.504 | 0.865 | 0.805 | 0.842 |
| | CIMSleepNet(Shen et al., 2024) | 0.849 | 0.799 | 0.797 | - | - | - | - | - |
| | cVAN(Yang et al., 2025) | 0.864 | 0.811 | 0.812 | 0.947 | 0.567 | 0.877 | 0.849 | 0.861 |
| | **S³Net** | **0.869** | **0.828** | **0.818** | **0.950** | **0.571** | **0.877** | **0.868** | **0.877** |

*Note: $\kappa$ denotes Cohen's kappa. The bold values indicate the best performance, and the underline values indicate the second-best.*

## 4.4 COMPARATIVE EXPERIMENT RESULTS

Through comprehensive evaluation across three benchmark datasets with 10-fold cross-validation and results averaged over three random seeds to mitigate randomness (Table 1), the proposed $S^3$Net model achieves SOTA performance in all evaluation metrics - overall accuracy, macro F1 score, and Cohen's Kappa ($\kappa$) (See Appendix A.4). On **ISRUC-S1**, $S^3$Net achieves 85.6% accuracy, outperforming cVAN (83.5%) and MixSleepNet (82.9%), with particularly significant gains in the challenging N1 stage (F1 score: 0.628). For **ISRUC-S3**, it attains 86.6% accuracy, surpassing cVAN

(85.6%) and StAGN (84.4%), while showing substantial improvements in N2 and N3 classification. On **Sleep-EDF-153**, S³Net reaches 86.9% accuracy, exceeding all compared methods including cVAN (86.4%), while maintaining balanced performance across all stages. Overall, these consistent improvements, particularly in transitional stages N1 and N2, validate the effectiveness of S³Net's stage-aware architecture and cross-domain feature integration strategy. Furthermore, stage-wise performance nuances are revealed through the confusion matrices in Figure 3. Although N1 achieves the highest F1 score as indicated in Table 1, it exhibits frequent misclassification as W and N2 across all datasets. This is likely attributable to high transition probabilities toward these stages. In contrast, the remaining stages demonstrate consistently high and stable classification performance.

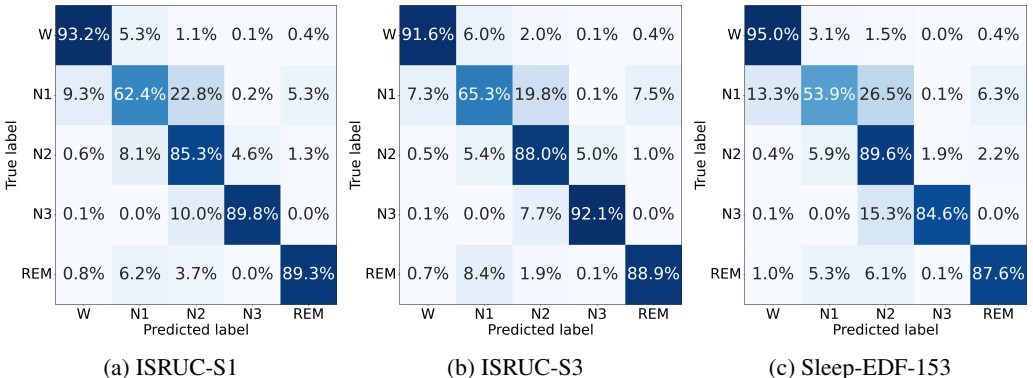

(a) ISRUC-S1        (b) ISRUC-S3        (c) Sleep-EDF-153

Figure 3: Confusion matrices of S³Net on three datasets: (a) ISRUC-S1, (b) ISRUC-S3, and (c) Sleep-EDF-153, aggregated over the 10-fold test sets.

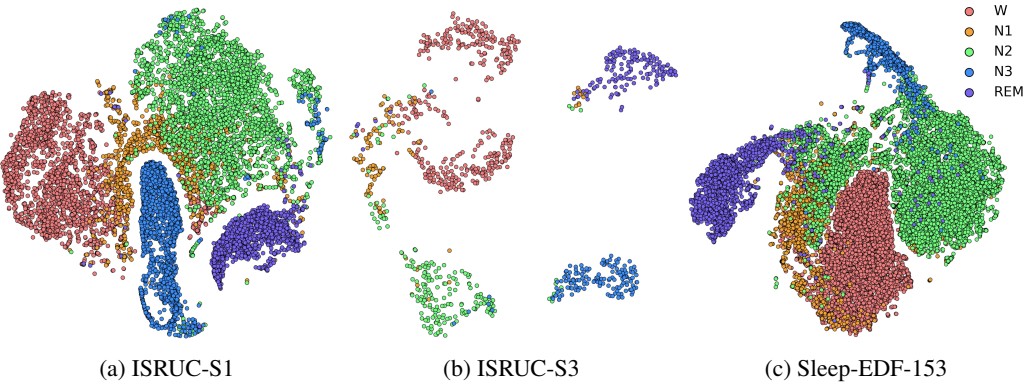

(a) ISRUC-S1        (b) ISRUC-S3        (c) Sleep-EDF-153

Figure 4: t-SNE visualization of the discriminative feature space learned by S³Net for the (a) ISRUC-S1, (b) ISRUC-S3, and (c) Sleep-EDF-153 datasets.

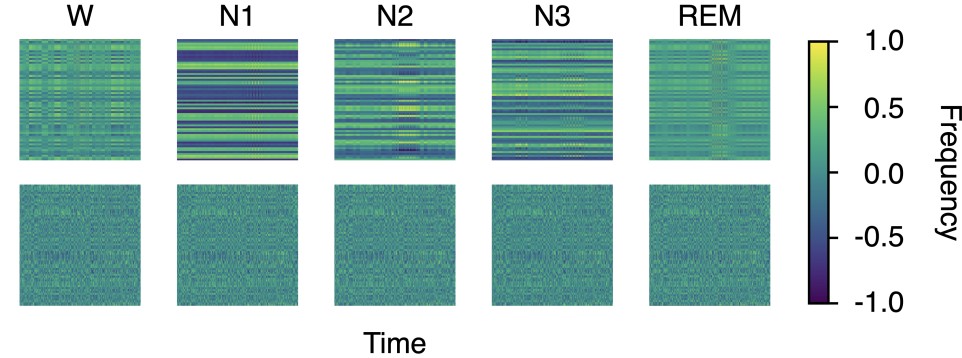

Figure 5: t-ALN visualization of $X_1^Q$ time–frequency spectrograms. Each spectrogram is plotted with time on the horizontal axis and frequency on the vertical axis.

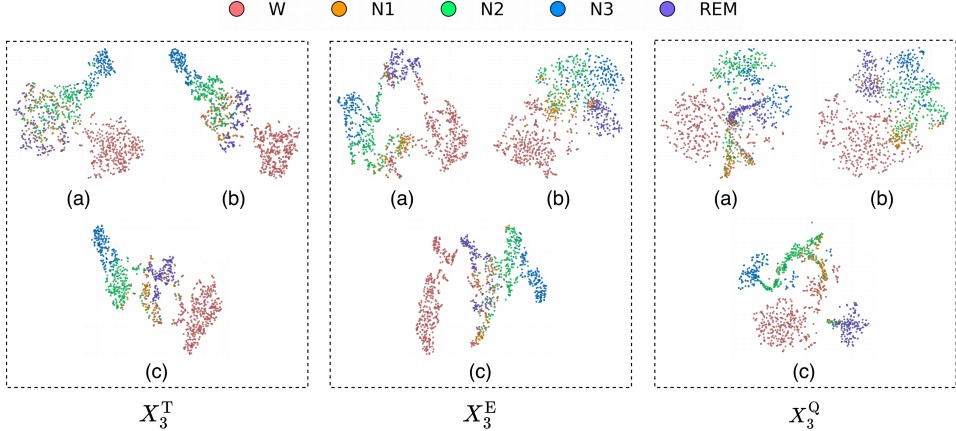

Figure 6: t-SNE visualizations of the CST output ($X_3^T$), energy representations ($X_3^E$), and temporally aligned features ($X_3^Q$) under different alignment configurations: (a) No Cross-Layer Interaction(no fusion between $X_1^Q$, $X_2^Q$ and $X_1^T$, $X_2^T$), (b)No-ALN, and (c) S³Net.

Table 2: Ablation study results of S³Net on the three datasets: ISRUC-S1, ISRUC-S3, and Sleep-EDF-153.

| Variant | t-ALN | SAE | $L_{aux}$ | ISRUC S1 | | | ISRUC S3 | | | Sleep-EDF-153 | | |
|---|---|---|---|---|---|---|---|---|---|---|---|---|
| | | | | Acc | F1 | $\kappa$ | Acc | F1 | $\kappa$ | Acc | F1 | $\kappa$ |
| M1 | × | × | × | 0.785 | 0.779 | 0.724 | 0.791 | 0.778 | 0.731 | 0.806 | 0.768 | 0.736 |
| M2 | ✓ | × | × | 0.809 | 0.803 | 0.754 | 0.818 | 0.800 | 0.767 | 0.822 | 0.778 | 0.756 |
| M3 | × | ✓ | × | 0.821 | 0.816 | 0.770 | 0.827 | 0.817 | 0.778 | 0.832 | 0.790 | 0.771 |
| M4 | × | ✓ | ✓ | 0.834 | 0.827 | 0.789 | 0.841 | 0.830 | 0.796 | 0.847 | 0.802 | 0.790 |
| M5 | ✓ | ✓ | × | 0.846 | 0.835 | 0.801 | 0.852 | 0.838 | 0.810 | 0.853 | 0.811 | 0.798 |
| M6 (S³Net) | ✓ | ✓ | ✓ | **0.856** | **0.842** | **0.814** | **0.866** | **0.855** | **0.827** | **0.869** | **0.828** | **0.818** |

*Note: $\kappa$ denotes Cohen's kappa. The bold values indicate the best performance, and the underline values indicate the second-best.*

## 4.5 ABLATION STUDY

To rigorously evaluate each core component of S³Net, we conduct an ablation study on all three datasets, ISRUC-S1, ISRUC-S3, and Sleep-EDF-153. As summarized in Table 2, the progressive performance improvements from variants M1 to M6 (S³Net) are highly consistent across datasets, systematically validating the efficacy of each proposed module. The baseline model (M1), without any specialized components, exhibits the lowest performance on all metrics. Introducing the t-ALN in M2 yields substantial gains across accuracy, macro F1, and Cohen's $\kappa$, confirming its critical role in bridging time- and frequency-domain representations for synergistic integration. Equipping the model with the SAE module in M3 and further adding the auxiliary loss $L_{aux}$ in M4 lead to additional, steady improvements, underscoring the importance of stage-specific feature refinement and enhanced discrimination. When t-ALN and SAE are jointly employed in M5, performance is markedly boosted and becomes the second-best configuration on the three datasets, highlighting the complementary benefits of cross-domain alignment and stage-aware modeling. Finally, the full M6 (S³Net), integrating t-ALN, SAE, and $L_{aux}$, consistently achieves the best accuracy, macro F1, and Cohen's $\kappa$ on ISRUC-S1, ISRUC-S3, and Sleep-EDF-153, demonstrating that the proposed design choices transfer robustly across heterogeneous cohorts and recording conditions.

## 4.6 VISUALIZATION AND INTERPRETABILITY

**Discriminative Feature Visualization.** Figure 4 presents the t-SNE visualization of S³Net's features, showing well-separated clusters for all five sleep stages across three datasets. The clear separation demonstrates the model's ability to learn highly discriminative representations, providing strong visual evidence of its robustness and generalization capacity for accurate sleep stage identification.

**Effect of t-ALN.** To intuitively demonstrate the functionality of the t-ALN component, time-frequency spectrograms of different sleep stages and alignment types are visualized. As shown in Figure 5, compared to the No-ALN, t-ALN captures more distinct and stage-specific patterns while exhibiting clearer structural separation. The t-SNE results in Figure 6 further confirm that S³Net achieves superior inter-class separation, validating the effectiveness of t-ALN in enhancing spectral-temporal alignment.

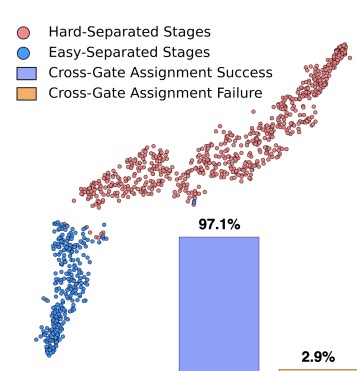

**Effect of Cross-Gate.** As shown in Figure 7, the t-SNE visualization clearly reveals a distinct separation between the different distinct and transitional stages when using the proposed cross-gate features. The corresponding bar chart illustrates a 97.1% cross-gate assignment success rate, further demonstrating the significant effectiveness of the proposed mechanism and auxiliary loss in enhancing the model's overall performance.

Figure 7: t-SNE visualization of the cross-gate features and corresponding bar charts.

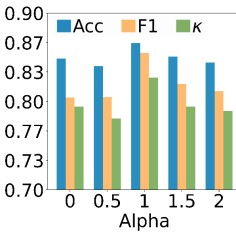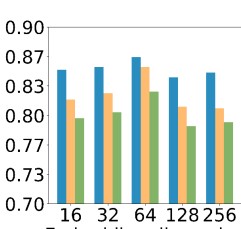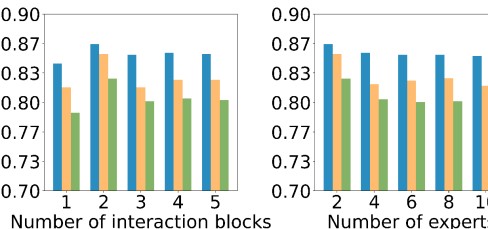

Figure 8: Hyperparameter performance of S³Net on ISRUC-S3.

**Hyperparameter Tuning.** Figure 8 shows that S³Net achieves the highest accuracy with an optimal configuration: $\alpha = 1$, the embedding dimension is set to 64, 2 interaction blocks (each comprising a Residual Block, t-ALN, and CST), and 2 experts. Performance varies noticeably as these hyperparameters change, with overly small or excessively large settings consistently leading to a decline across Acc, F1, and $\kappa$ scores. These results indicate that balanced auxiliary supervision, moderate embedding capacity, and a compact model structure collectively contribute to improved generalization performance.

## 5 CONCLUSION

In this paper, we have presented S³Net, a novel deep learning framework designed to address two fundamental challenges in automated sleep staging: the difficulty in distinguishing transitional sleep stages (N1 and N2) and the ineffective fusion of time-frequency representations. Our proposed SAE module introduces a structured approach to sleep stage classification by explicitly separating stages into distinct complexity groups and processing them through specialized network branches. This design allows for targeted feature refinement, significantly improving performance on ambiguous transitional stages. Complementing this, our t-ALN module enables coherent cross-domain integration by projecting frequency-derived features onto the temporal axis, effectively bridging the representation gap between time and frequency domains. Extensive experiments on three public datasets demonstrate that S³Net consistently achieves state-of-the-art performance, with particular improvement in classifying the challenging N1 and N2 stages. Beyond quantitative improvements, our framework offers valuable insights into sleep architecture through its interpretable design. The proposed methodology not only advances automated sleep staging performance but also provides a principled approach for handling heterogeneous complexity in physiological signal analysis.

## 6 ETHICS STATEMENT

The datasets used in this paper are sourced from publicly available datasets. All data were used in accordance with their original licenses and intended purposes for academic research. As the data are public and do not contain personally identifiable information, this study did not require ethics approval.

## 7 REPRODUCIBILITY STATEMENT

The model architecture is introduced in detail with equations and figures in the main text. All the implementation details are included in the Appendix A.5, including dataset descriptions, metrics of each task, model configurations, and experiment settings. Code is available at this repository: https://anonymous.4open.science/r/S3Net/

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

## A APPENDIX

### A.1 USAGE OF LLMS

We used large language models (LLMs) solely for polishing the writing, such as improving sentence fluency and fixing grammar issues. No content generation, idea creation, or experimental analysis was conducted by LLMs.

## A.2 ALGORITHM OF T-ALN

---

**Algorithm 1** t-ALN

---

**Require:** Spectral feature $X \in \mathbb{R}^{B \times C_s \times F_s \times T_s}$.
**Ensure:** Aligned sequence $Y \in \mathbb{R}^{B \times L_{out} \times D_{model}}$.
   $X^E \leftarrow X \odot X$ {Element-wise square to get energy}
   $W^F \leftarrow \text{Softmax}(\text{Mean}(X^E, \text{dims} = (C_s, T_s)))$ {Compute frequency-wise weights}
   $W^C \leftarrow \text{Softmax}(\text{Mean}(X^E, \text{dims} = (F_s, T_s)))$ {Compute channel-wise weights}
   $\tilde{X}^E \leftarrow X^E \cdot W^F \cdot W^C$ {Modulate energy with computed weights}
   $X^W \leftarrow \text{Linear}(\text{PermuteToFeatureLast}(\tilde{X}^E))$ {Project to $D_s$ dim; $X^W \in \mathbb{R}^{B \times T_s \times F_s \times D_s}$}
   $\tilde{X}^W \leftarrow X^W + \text{PositionalEncoding}(\tilde{X}^E)$ {Add positional encoding}
   $\tilde{X}^W \leftarrow \text{ReshapeToSequence}(\tilde{X}^W)$ {Form sequence $\tilde{X}^W \in \mathbb{R}^{B \times (T_s \times D_s) \times F_s}$}
   $S_{attn} \leftarrow \text{SelfAttention}(\tilde{X}^W)$ {Apply self-attention on the sequence}
   $Y \leftarrow \text{Linear}(S_{attn})$ {Final projection to $D_{model}$; $Y \in \mathbb{R}^{B \times (T_s \times D_s) \times D_{model}}$}
   **return** $Y$

---

The t-ALN module aligns spectral and temporal features by first refining spectral energy maps with frequency- and channel-wise attention weights. These refined features are then passed through a Time-Window-Wise Linear layer, followed by a Transformer layer to capture temporal and spectral dependencies. The output query representation is used in Cross-Swin Transformer with the temporal features, enhancing sleep stage classification by suppressing noise and preserving discriminative patterns.

## A.3 CST IMPLEMENTATION DETAILS

CST performs window-based cross-attention to fuse temporal features $X_i^T \in \mathbb{R}^{B \times L_i \times D_i}$ with frequency-aligned queries $X_i^Q \in \mathbb{R}^{B \times L_i \times D_i}$ produced by t-ALN from energy maps $X_i^E$. Prior to attention, both streams undergo identical pre-processing: layer normalization, right padding to a multiple of the window length $W_i$, cyclic shift $s_i \in \{0, \lfloor W_i/2 \rfloor\}$, and partition into non-overlapping windows of size $W_i$. The deepest aligned feature $X_3^Q$ is consumed by SAE for routing and by expert heads. The aligned spectral representations $X_1^Q, X_2^Q, X_3^Q$ are preprocessed identically to $X_i^T$ before window attention (layer normalization, right padding to a multiple of the window length $W_i$, cyclic shift, and window partition), ensuring shared positional indexing and relative-position bias tables across streams. Within a window $w$, multi-head cross-attention uses external queries from $X_i^Q(w)$ and keys/values from $X_i^T(w)$:

$$Q = X_i^Q(w)\,W_q, \quad K = X_i^T(w)\,W_k, \quad V = X_i^T(w)\,W_v, \quad d_h = D_i/H,$$

with $H$ heads and per-head scale $d_h$. The attention weights are

$$A = \text{softmax}\left(\frac{QK^\top}{\sqrt{d_h}} + B_{\text{rel}} + M_w\right),$$

where $B_{\text{rel}}$ is the 1D relative-position bias indexed by window offsets and $M_w$ is the shifted-window mask. The output is

$$O = A\,V\,W_o,$$

merged back to the sequence, followed by residual addition, layer normalization, an MLP, and an Efficient Channel Attention (ECA) block. Blocks are stacked in BasicLayer with alternating regular and shifted windows; masks $M_w$ are cached per $(L_i, W_i, s_i)$. Let $\tilde{X}_i^T$ denote the output features from the stacked CST blocks. PatchMerging is then applied to halve sequence length and double channels:

$$X_{i+1}^T = \text{PatchMerging}(\tilde{X}_i^T), \quad L_{i+1} = \lceil L_i/2 \rceil, \ D_{i+1} = 2D_i.$$

Expert CST blocks do not employ PatchMerging. Progressive fusion proceeds as $(X_1^T, X_1^Q) \rightarrow X_2^T$ and $(X_2^T, X_2^Q) \rightarrow X_3^T$; $X_3^Q$ interfaces with SAE gating and experts. Per block, the dominant cost scales with the number of windows $n_i = \lceil L_i/W_i \rceil$ as $\mathcal{O}(B\,n_i\,H\,W_i^2)$; hierarchical merging reduces $L_i$ and thus attention cost in deeper layers.

## A.4 EVALUATION METRICS

We report three standard classification metrics to evaluate model performance: accuracy, F1 score, and Cohen's kappa coefficient.

- **Overall accuracy** is the proportion of correctly predicted segments over the entire aggregated test set across all folds.
- **F1 score** is the harmonic mean of precision and recall. We report the macro F1 over all five sleep stages, treating each class equally regardless of class frequency. In addition, we also report the per-stage F1 score (W, N1, N2, N3, REM), which are computed separately for each class and reflect stage-specific performance.
- **Cohen's kappa** evaluates inter-rater agreement normalized by chance, and is defined as

$$\kappa = \frac{p_o - p_e}{1 - p_e} \tag{7}$$

where $p_o$ is the observed accuracy and $p_e$ is the expected accuracy by random chance. It provides a robust measure that accounts for label imbalance.

All metrics are computed per dataset on the test set using standard sklearn implementations.

## A.5 HYPERPARAMETER SETTING

We train all models using the AdamW optimizer with a learning rate of 3e-4 and a batch size of 32. The regularization weight $\alpha$, embedding dimension, and number of interaction blocks (Residual Block, Time Alignment and Cross Swin Transformer) are chosen based on validation performance, while the number of experts is fixed to one for both hard- and easy-separated stages. A complete list of hyperparameters is summarized in Table 3.

Table 3: Hyper-parameter Settings

| Hyper-parameter | Value |
|---|---|
| Optimizer | AdamW |
| Learning rate | 3e-4 |
| Batch-size | 32 |
| $L_{aux}$ regularization weight ($\alpha$) | 1.0 |
| Embedding dimension | 64 |
| Number of interaction block | 2 |
| Number of experts (Hard-Separated) | 1 |
| Number of experts (Easy-Separated) | 1 |
| Epoch | 30 |

## A.6 MODEL COMPLEXITY ANALYSIS

Our S³Net model comprises 6.49 million parameters and requires 2.70 GFLOPs to process a single 30-second sleep segment, where FLOPs are estimated as $2 \times$ MACs. As shown in Table 4, S³Net uses substantially fewer parameters than MVF-SleepNet (39.51M) and slightly fewer than cVAN (7.58M), while achieving considerably lower latency during both training (75.69 ms) and inference (24.02 ms). All latency measurements were obtained with a batch size of 32 and averaged over 1,000 runs. Although S³Net requires more FLOPs than cVAN, the resulting computational overhead remains moderate and still leads to noticeably faster inference in practice. These results indicate that S³Net achieves a favorable balance between model capacity and computational efficiency.

## A.7 DATASETS

We evaluate the proposed S³Net on three publicly available polysomnographic (PSG) datasets: ISRUC-S1, ISRUC-S3, and Sleep-EDF-153, which is also referred to as Sleep-EDF-78. All datasets

Table 4: Model efficiency comparison.

| Model | Params (M) | GFLOPs | Train (ms) | Infer (ms) |
|---|---|---|---|---|
| MVF-SleepNet | 39.51 | 11.06 | – | – |
| cVAN | 7.58 | 0.47 | 157.01 | 66.72 |
| S$^3$Net | 6.49 | 2.70 | 75.69 | 24.02 |

Table 5: The General Description of Sleep Datasets

| Dataset | Subject | Segments | W | N1 | N2 | N3 | REM |
|---|---|---|---|---|---|---|---|
| ISRUC-S1 | 100 | 87,187 | 20,098 | 11,062 | 27,511 | 17,251 | 11,265 |
| ISRUC-S3 | 10 | 8,589 | 1,674 | 1,217 | 2,616 | 2,016 | 1,066 |
| Sleep-EDF-153 | 153 | 195,292 | 65,790 | 21,522 | 69,106 | 13,039 | 25,835 |

contain full-night sleep recordings segmented into 30-second segments, with stage annotations provided by certified experts following the AASM or R&K standards. Each segment is labeled as one of five sleep stages: Wake (W), N1, N2, N3, and REM. A summary of dataset statistics, including the number of subjects and stage-wise segment distributions, is provided in Table 5.

- **ISRUC-S1** (Khalighi et al., 2016) includes overnight PSG recordings from 100 subjects (55 males and 45 females) with various sleep disorders, aged between 20 and 85 years. Each recording lasts approximately eight hours and contains 12 PSG channels: 6 EEG (C3-A2, C4-A1, F3-A2, F4-A1, O1-A2, O2-A1), 3 EMG (chin, leg-1, leg-2), 2 EOG, and 1 ECG. For our experiments, we select a 10-channel subset composed of 6 EEG, 2 EOG, 1 EMG (chin), and 1 ECG signal. The data are downsampled from 200 Hz to 100 Hz for consistency across datasets. After segmentation, a total of 87,187 30-second segments are obtained for training and evaluation.

- **ISRUC-S3** (Khalighi et al., 2016) follows the same data acquisition protocol and channel configuration as ISRUC-S1 but focuses on healthy individuals. It contains PSG recordings from 10 healthy subjects (9 males and 1 female), aged between 30 and 58 years. Using the same preprocessing pipeline and 10-channel subset as in ISRUC-S1 (6 EEG, 2 EOG, 1 chin EMG, and 1 ECG, downsampled from 200 Hz to 100 Hz), we obtain 8,589 annotated 30-second segments. This dataset is particularly useful for validating generalization performance on non-pathological data.

- **Sleep-EDF-153** (Kemp et al., 2000) is derived from the expanded Sleep-EDF dataset and contains PSG recordings from 153 healthy subjects aged 25 to 101 years. Each recording is sampled at 100 Hz and annotated according to the older R&K standard. We select four PSG channels for input: two EEG (Fpz-Cz and Pz-Oz), one EOG, and one EMG. For our experiments, we select a 2-channel subset composed of 2 EEG. This dataset yields a total of 195,292 labeled segments after segmentation and preprocessing. Compared to ISRUC datasets, Sleep-EDF offers a larger subject pool but fewer input channels.

To ensure consistent input dimensionality and leverage both spatial and spectral diversity, we construct input sequences using the available EEG, EOG, EMG, and ECG signals in each dataset, resulting in 10-channel inputs for ISRUC and 2-channel(EEG) inputs for Sleep-EDF. All datasets are used with 10-fold subject-wise cross-validation, ensuring no subject leakage between training and testing folds.

### A.8 BASELINES

To comprehensively evaluate S$^3$Net, we compare it against a broad set of baselines spanning five major categories. Temporal models such as DeepSleepNet (Supratak et al., 2017), TinySleepNet (Supratak & Guo, 2020), XSleepNet (Phan et al., 2022), and SleePyCo (Lee et al., 2024) adopt sequential architectures that operate on raw EEG or frame-level features, focusing primarily on temporal dependencies without explicitly modeling spectral information. Frequency-aware methods including SeqSleepNet (Phan et al., 2019), MVF-SleepNet (Li et al., 2022), and cVAN (Yang et al., 2025) leverage spectrograms, filter banks, or energy-based representations to jointly encode

temporal and spectral dynamics, often improving performance on stages with prominent frequency signatures such as N3 and REM. Graph-based approaches such as STGCN (Jia et al., 2020b), MST-GCN (Jia et al., 2020a), DGraphormer-SleepNet (Huang et al., 2025) and StAGN (Chen et al., 2023a) explicitly capture spatial dependencies across EEG channels using static or learnable graph structures, enhancing inter-channel relational modeling. Transformer-based, attention-augmented, and probabilistic models such as MixSleepNet (Ji et al., 2024) and BSTT (Liu & Jia, 2023) incorporate global attention mechanisms or uncertainty estimation to facilitate long-range dependency modeling and robust decision making under ambiguity. Finally, multimodal and contrastive learning methods including SLEEPSMC (Ma et al., 2025) and CIMSleepNet (Shen et al., 2024) aim to improve robustness under modality corruption or incompleteness, often through cross-view supervision or modality-invariant learning objectives. Collectively, these baselines form a comprehensive benchmark that encompasses frequency-aware and frequency-blind designs, as well as sequential, spatial, and multimodal modeling paradigms.

## A.9 Visualization of Sleep Staging Hypnograms

Across the 10-fold cross-validation on the ISRUC-S3 dataset, our model consistently demonstrates strong performance across all five sleep stages (W, N1, N2, N3, REM). As shown in the hypnogram comparisons in Figure 12, the predicted sleep stage sequences closely align with the ground truth, indicating that the model effectively captures the temporal structure and transitions of sleep. Notably, even for the more challenging transitional stages such as N1 and N2, the model maintains robust predictive accuracy, highlighting its stability and generalization ability across folds. Among these, recording 7 in Figure 12h stands out as the best-performing fold, with the predicted hypnogram almost perfectly matching the ground truth throughout the entire sleep cycle; this exceptional result is partially attributed to the relatively low proportion of N1 epochs in this fold, which reduces ambiguity and confusion during classification. Since N1 often overlaps with adjacent stages such as W and N2, folds with a higher presence of N1 are more susceptible to misclassifications, whereas the reduced presence in recording 7 allows the model to generate cleaner and more temporally coherent predictions.

Table 6: Generalization on WESAD(Schmidt et al., 2018) (PPG, 4-class).

| Model | Acc | AUROC | AUPRC | F1 | $\kappa$ |
|---|---|---|---|---|---|
| REBAR(Xu et al., 2024) | 0.418 | 0.698 | 0.446 | – | – |
| cVAN(Yang et al., 2025) | 0.691 | 0.855 | 0.710 | 0.664 | 0.572 |
| ResNet(He et al., 2015) | 0.713 | 0.877 | 0.746 | 0.682 | 0.593 |
| HuBERT(Narain et al., 2025) | 0.775 | 0.820 | – | – | – |
| S$^3$Net | 0.853 | 0.911 | 0.833 | 0.819 | 0.787 |

*Note: $\kappa$ denotes Cohen's kappa. The bold values indicate the best performance, and the underline values indicate the second-best.*

## A.10 Generalization to PPG Signals

To examine whether the proposed architecture generalizes to physiological modalities beyond PSG, we further evaluate it on the publicly available WESAD (Schmidt et al., 2018) dataset using single-lead PPG signals. Unlike the multi-channel PSG recordings used in the main experiments (including EEG, EOG, EMG, and ECG), the WESAD (Schmidt et al., 2018) dataset offers a fundamentally different sensing modality based on wrist-worn PPG (blood-volume pulse, BVP). WESAD is a multimodal physiological dataset collected from 15 subjects using a chest-worn RespiBAN device and a wrist-worn Empatica E4, providing signals such as ECG, EDA, respiration, temperature, acceleration, and BVP. In this study, we exclusively use the single-channel BVP signal recorded by the Empatica E4 at 64 Hz. Following common preprocessing practices, the BVP stream is segmented into non-overlapping 1-minute windows (3,840 samples each), yielding 1,305 segments across all subjects, of which 666 are annotated with affective labels.

From the labeled BVP segments, we define a four-class classification task consisting of baseline, stress, amusement, and meditation. The class distribution is approximately baseline (42.7%), stress

(24.0%), amusement (12.4%), and meditation (20.9%). The BVP signal reflects pulse morphology and heart rate dynamics, making it suitable for single-modality affect and stress recognition. Because the original SAE module in our framework includes sleep-stage-specific experts tailored to PSG-based sleep staging, we remove these experts when evaluating on WESAD. The remaining cross-gating module is retained, and the output layer is replaced with a generic four-class classification head appropriate for the BVP-based task. As summarized in Table 6, S³Net markedly outperforms representative baselines such as REBAR (Xu et al., 2024), cVAN (Yang et al., 2025), ResNet (He et al., 2015), and HuBERT (Narain et al., 2025) across accuracy, AUROC, AUPRC, F1, and Cohen's $\kappa$, indicating that the proposed architecture generalizes well to single-lead PPG for affective state recognition.

## A.11  CROSS-DATASET TRANSFERABILITY

To further evaluate the generalization capability of the proposed framework, we conduct cross-dataset experiments by training on ISRUC-S1 and testing on ISRUC-S3. This setup reflects a realistic deployment scenario where a model trained on one cohort must be applied to data collected under different conditions, including variations in subject populations, recording environments, and sensor characteristics. Cross-dataset sleep staging is substantially more challenging than within-dataset evaluation because the distribution shift between datasets often leads to degraded performance, especially for models that rely on dataset-specific signal patterns.

Table 7 summarizes the cross-dataset results. Overall, all baseline models suffer noticeable performance drops compared to their within-dataset results, confirming the difficulty of cross-dataset generalization. Among the baselines, cVAN achieves an accuracy of 0.826 and F1 of 0.807, outperforming StAGN and MVF-SleepNet. However, S³Net exhibits the strongest robustness under this distribution shift, reaching 0.834 accuracy, 0.816 F1, and a substantial $\kappa$ of 0.830. These results demonstrate that the proposed architecture not only captures discriminative intra-dataset patterns but also effectively generalizes to previously unseen data distributions. The superior cross-dataset performance highlights the model's potential for real-world clinical deployment, where training and test data are rarely perfectly aligned.

Table 7: Cross-dataset sleep staging performance (Train: ISRUC-S1; Test: ISRUC-S3). Higher is better.

| Train | Test | Model | Acc | F1 | $\kappa$ |
|---|---|---|---|---|---|
| ISRUC-S1 | ISRUC-S3 | StAGN(Chen et al., 2023a) | 0.795 | 0.779 | – |
| | | MVF-SleepNet(Li et al., 2022) | 0.800 | 0.788 | – |
| | | cVAN(Yang et al., 2025) | 0.826 | 0.807 | – |
| | | S³Net | 0.834 | 0.816 | 0.830 |

## A.12  SINGLE-MODALITY COMPARISON

Beyond full-modality evaluation, we further analyze the proposed framework under single-modality settings to assess its robustness when only one PSG modality is available. Specifically, we consider three input configurations: EEG-only, EOG-only, and EMG-only, while keeping the network architecture and training protocol unchanged except for the input channel dimension. All experiments follow the same subject-wise cross-validation protocol as in the main study. Single-modality sleep staging is challenging because each physiological signal provides only partial information: EEG contains the most discriminative oscillatory patterns, EOG captures eye movements characteristic of REM, and EMG mainly reflects muscle-tone changes, leading to different levels of class separability across modalities.

Table 8 reports the single-modality results on ISRUC-S3. As expected, EEG yields the best performance due to its rich stage-specific cues, EOG achieves moderate performance, and EMG performs the worst given its limited coverage of sleep-related dynamics. S³Net consistently outperforms all baseline models under every single-modality setting. With EEG-only input, it achieves 0.8454 accuracy. Under EOG-only input, it reaches 0.8198 accuracy, showing strong robustness even without EEG. For EMG-only input, S³Net still obtains the highest scores, with 0.5857 accuracy, indicating that the architecture remains effective even when provided with severely limited physiological cues.

Table 8: Single-modality sleep staging comparison (EEG/EOG/EMG) on ISRUC-S3. Higher is better.

| Modality | Method | Overall results | | | F1 for each category | | | | |
|---|---|---|---|---|---|---|---|---|---|
| | | Acc | F1 | $\kappa$ | Wake | N1 | N2 | N3 | REM |
| EEG | AttnSleep(Eldele et al., 2021) | 0.7338 | 0.7105 | 0.6592 | 0.8581 | 0.4636 | 0.7320 | 0.8524 | 0.6463 |
| | DAN(Tang et al., 2022) | 0.7212 | 0.6791 | 0.6400 | 0.8077 | 0.3511 | 0.7352 | 0.8686 | 0.6328 |
| | BSTT(Liu & Jia, 2023) | 0.7191 | 0.6921 | 0.6371 | 0.8061 | 0.4312 | 0.6989 | 0.8502 | 0.6742 |
| | XSleepNet(Phan et al., 2022) | 0.6555 | 0.6322 | 0.5614 | 0.8525 | 0.4562 | 0.6225 | 0.8015 | 0.4281 |
| | SleepPrintNet(Jia et al., 2020c) | 0.5459 | 0.4862 | 0.3924 | 0.5109 | 0.3404 | 0.6161 | 0.6669 | 0.2968 |
| | MMASleepNet(Yubo et al., 2022) | 0.6313 | 0.5975 | 0.5150 | 0.7815 | 0.3486 | 0.6771 | 0.6471 | 0.5333 |
| | SimCLR(Chen et al., 2020) | 0.7338 | 0.7163 | 0.6598 | 0.8777 | 0.4978 | 0.6883 | 0.8260 | 0.6915 |
| | DrFuse(Yao et al., 2024) | 0.7532 | 0.7138 | 0.6818 | 0.8780 | 0.3872 | 0.7794 | 0.8609 | 0.6636 |
| | MERL(Liu et al., 2024) | 0.7467 | 0.7295 | 0.6758 | 0.8524 | 0.5212 | 0.7328 | 0.8603 | 0.6808 |
| | SleepSMC(Ma et al., 2025) | 0.7646 | 0.7397 | 0.6969 | 0.8882 | 0.5069 | 0.7467 | 0.8636 | 0.6932 |
| | $S^3$Net | **0.8454** | **0.8303** | **0.8010** | **0.9124** | **0.6406** | **0.8456** | **0.9141** | **0.8309** |
| EOG | AttnSleep(Eldele et al., 2021) | 0.7226 | 0.6992 | 0.6416 | 0.8248 | 0.4608 | 0.7115 | 0.8591 | 0.6399 |
| | DAN(Tang et al., 2022) | 0.7136 | 0.6647 | 0.6288 | 0.7733 | 0.2902 | 0.7406 | 0.8652 | 0.6542 |
| | BSTT(Liu & Jia, 2023) | 0.4700 | 0.3163 | 0.2790 | 0.1169 | 0.2352 | 0.5895 | 0.6400 | 0.0000 |
| | XSleepNet(Phan et al., 2022) | 0.6288 | 0.6071 | 0.5233 | 0.6958 | 0.3684 | 0.6572 | 0.7882 | 0.5260 |
| | SleepPrintNet(Jia et al., 2020c) | 0.3745 | 0.2531 | 0.1788 | 0.3553 | 0.0239 | 0.5680 | 0.0000 | 0.3183 |
| | MMASleepNet(Yubo et al., 2022) | 0.2096 | 0.1745 | 0.0619 | 0.2750 | 0.2712 | 0.0000 | 0.0000 | 0.3264 |
| | SimCLR(Chen et al., 2020) | 0.7246 | 0.7007 | 0.6458 | 0.8097 | 0.4788 | 0.7096 | 0.8523 | 0.6529 |
| | DrFuse(Yao et al., 2024) | 0.6947 | 0.6799 | 0.6078 | 0.7522 | 0.4579 | 0.7115 | 0.8317 | 0.6460 |
| | MERL(Liu et al., 2024) | 0.6976 | 0.6741 | 0.6132 | 0.7996 | 0.3912 | 0.6808 | 0.8351 | 0.6640 |
| | SleepSMC(Ma et al., 2025) | 0.7444 | 0.7168 | 0.6697 | 0.8386 | 0.4765 | 0.7360 | 0.8722 | 0.6607 |
| | $S^3$Net | **0.8198** | **0.8006** | **0.7676** | **0.8959** | **0.5686** | **0.8153** | **0.9067** | **0.8168** |
| EMG | AttnSleep(Eldele et al., 2021) | 0.3915 | 0.3814 | 0.2191 | 0.5096 | 0.2067 | 0.3804 | 0.4152 | 0.3950 |
| | DAN(Tang et al., 2022) | 0.4048 | 0.3381 | 0.2267 | 0.5541 | 0.0065 | 0.4670 | 0.2262 | 0.4365 |
| | BSTT(Liu & Jia, 2023) | 0.3046 | 0.0934 | 0.0000 | 0.0000 | 0.0000 | 0.4669 | 0.0000 | 0.0000 |
| | XSleepNet(Phan et al., 2022) | 0.3660 | 0.3484 | 0.1935 | 0.4519 | 0.1654 | 0.3665 | 0.3833 | 0.3748 |
| | SleepPrintNet(Jia et al., 2020c) | 0.3319 | 0.2313 | 0.0939 | 0.4214 | 0.0359 | 0.4327 | 0.0000 | 0.2667 |
| | MMASleepNet(Yubo et al., 2022) | 0.2517 | 0.1969 | 0.1062 | 0.4155 | 0.1450 | 0.0000 | 0.0000 | 0.4240 |
| | SimCLR(Chen et al., 2020) | 0.4177 | 0.3906 | 0.2435 | 0.5605 | 0.1397 | 0.4303 | 0.4258 | 0.3968 |
| | DrFuse(Yao et al., 2024) | 0.3857 | 0.3789 | 0.2318 | 0.6026 | 0.1923 | 0.3549 | 0.3272 | 0.4176 |
| | MERL(Liu et al., 2024) | 0.3981 | 0.3907 | 0.2348 | 0.4875 | 0.2077 | 0.3879 | 0.4008 | 0.4696 |
| | SleepSMC(Ma et al., 2025) | 0.4384 | 0.4075 | 0.2693 | 0.5868 | 0.1281 | 0.4404 | 0.4301 | 0.4523 |
| | $S^3$Net | **0.5857** | **0.5547** | **0.4609** | **0.7211** | **0.2623** | **0.5736** | **0.5968** | **0.6198** |

*Note: $\kappa$ denotes Cohen's kappa. The bold values indicate the best performance, and the underline values indicate the second-best.*

### A.13 Expert Group Comparison

To validate the design of our Stage-Aware Experts, we evaluated several alternative ways of grouping stages into experts on ISRUC-S3 (Table 9) to assess whether the hard-separated group (W, N1, N2) and easy-separated group (N3, REM) strategy performs best. Splitting N1 from the remaining stages yields a low accuracy of 0.848. Assigning Wake to one expert, a middle group comprising N1, N2, and N3 to another, and reserving a dedicated expert for REM yields only a slight gain of 0.851. Using a single expert for all non-REM stages and a dedicated expert for REM achieves 0.853. Combining Wake with N1 while merging N2, N3, and REM improves further to 0.856. Separating Wake, pairing N1 with N2, and isolating N3 with REM yields 0.858. Coupling Wake with REM while routing N1, N2, and N3 together performs relatively well at 0.860. In contrast, directing the hard-separated group comprising Wake, N1, and N2 to one expert and the easy-separated group comprising N3 and REM to another attains an accuracy of 0.866 on ISRUC-S3.

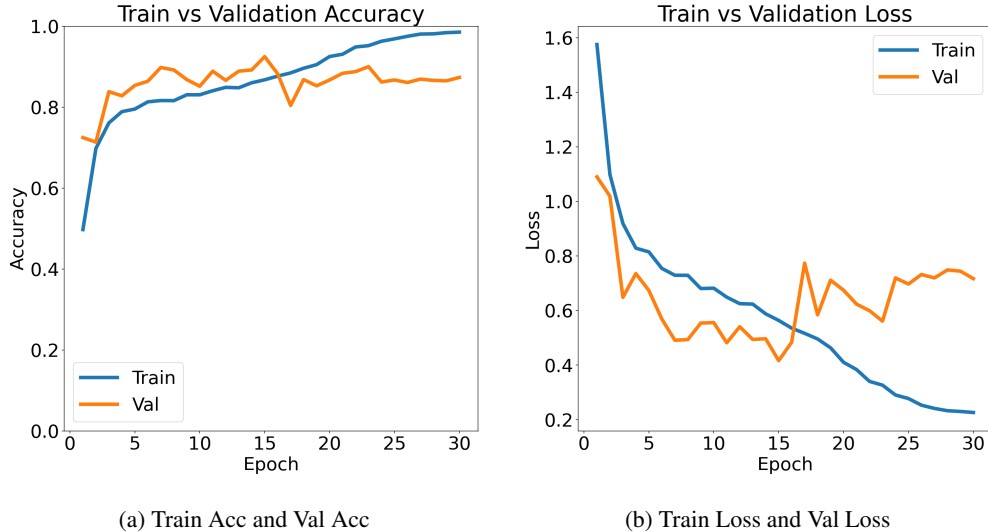

(a) Train Acc and Val Acc

(b) Train Loss and Val Loss

Figure 9: Training and validation loss and accuracy of the proposed model.

Table 9: Expert group comparison on ISRUC-S3. Higher is better.

| Group | Overall results | | | F1 for each category | | | | |
|---|---|---|---|---|---|---|---|---|
| | Acc | F1 | $\kappa$ | Wake | N1 | N2 | N3 | REM |
| (W,N2,N3,REM) vs (N1) | 0.848 | 0.835 | 0.804 | 0.909 | 0.639 | 0.839 | 0.916 | 0.877 |
| (W) vs (N1,N2,N3) vs (REM) | 0.851 | 0.839 | 0.808 | 0.908 | 0.648 | 0.842 | 0.913 | 0.884 |
| (W,N1,N2,N3) vs (REM) | 0.853 | 0.841 | 0.810 | 0.909 | 0.652 | 0.843 | 0.914 | 0.887 |
| (W,N1) vs (N2,N3,REM) | 0.856 | 0.844 | 0.814 | 0.920 | 0.656 | 0.848 | 0.917 | 0.877 |
| (W) vs (N1,N2) vs (N3,REM) | 0.858 | 0.847 | 0.817 | 0.914 | 0.661 | 0.852 | 0.920 | 0.885 |
| (W,REM) vs (N1,N2,N3) | 0.860 | 0.849 | 0.820 | 0.918 | 0.666 | 0.854 | 0.921 | 0.885 |
| (W,N1,N2) vs (N3,REM) (S³Net) | **0.866** | **0.855** | **0.827** | **0.924** | **0.678** | **0.858** | **0.927** | **0.887** |

*Note: $\kappa$ denotes Cohen's kappa. The bold values indicate the best performance, and the underline values indicate the second-best.*

### A.14 TRAINING DYNAMICS

Figure 9 summarizes the optimization dynamics of the model. The training loss decreases rapidly in the first few epochs and then gradually flattens, indicating stable convergence under the chosen learning rate schedule. On the validation set, accuracy increases steadily and reaches its peak around epochs 12 to 15, where the validation loss also attains its minimum. The consistency between the loss and accuracy trends suggests that the model maintains stable generalization during most of training.

In the early to mid stage of training, the validation loss is slightly lower than the training loss. This behavior is expected because the training objective includes explicit regularization terms such as weight decay and implicit stochastic regularization mechanisms such as dropout, batch normalization with minibatch statistics, and data augmentation. These components increase the training loss, whereas validation is computed in inference mode without such perturbations. As training proceeds, the model fits the training distribution more closely, the training loss continues to decrease, and a standard generalization gap emerges. Unless otherwise stated, all reported results are obtained from the checkpoint with the best validation performance, which in this run occurs at epoch 15.

### A.15 EFFECT OF THE AUXILIARY LOSS WEIGHT

We analyze the effect of the auxiliary loss weight $\alpha$ by visualizing the optimization dynamics in Figure 10, based on the total loss $L_{\text{total}}$ defined in Eq. equation 6. For each value of $\alpha$, we project the

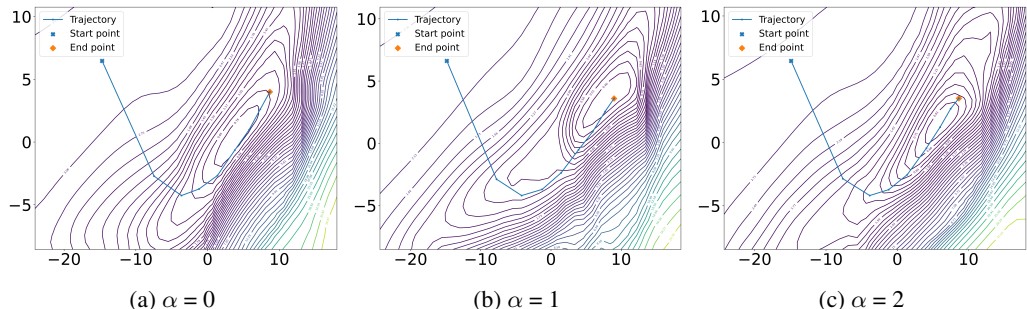

Figure 10: Projected loss contours and optimization trajectories for different values of $\alpha$.

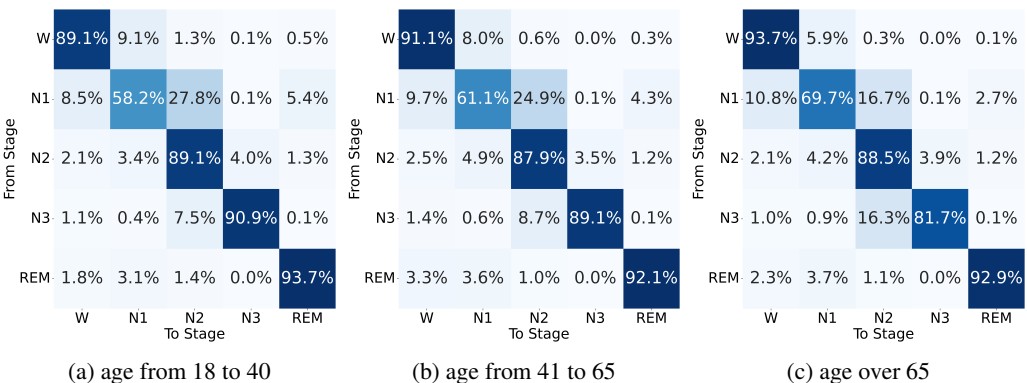

Figure 11: Stage transition probability matrices pooled over ISRUC-S3, ISRUC-S1, and Sleep-EDF-153, stratified by age group.

loss surface onto a two-dimensional subspace and plot the contour lines together with the gradient-based trajectory from a shared initialization.

When $\alpha = 0$, the auxiliary loss is disabled and the trajectory in Figure 10a crosses the main valley and settles in a shallow side basin rather than at the deepest region of the landscape, indicating that the optimization tends to converge to a suboptimal local minimum. With $\alpha = 1$, as shown in Figure 10b, the trajectory follows a smooth path into the central basin and converges near the lowest point of the projected loss surface; the updates are stable and well aligned with the descent directions, suggesting that this auxiliary weight regularizes the landscape and guides training towards a better minimum. Increasing the weight to $\alpha = 2$ yields the trajectory in Figure 10c, where the path again bypasses the deepest region and ends in a higher-loss basin, implying that an overly large auxiliary weight distorts the main objective and leads to another suboptimal optimum. Overall, $\alpha = 1$ provides the best trade-off, enabling convergence to the lowest basin in the projected landscape and yielding the strongest empirical performance, and is therefore adopted in our main experiments.

### A.16 AGE-STRATIFIED STAGE TRANSITION DYNAMICS

To assess whether the hard/easy stage partition is stable across age and across datasets, we investigate subjects from ISRUC-S3, ISRUC-S1, and Sleep-EDF-153, stratify them into three age groups, and recompute the stage transition matrices, as shown in Figure 11. Across all groups (ages 18 to 40, 41 to 65, and over 65 years), the same qualitative block structure is preserved: W, N1, and N2 exhibit elevated transition probabilities among one another, while N3 and REM are dominated by self-transitions with no strong preference toward any single other stage. Although individual transition probabilities vary moderately with age, these changes do not alter the underlying separation between the {W, N1, N2} group and the {N3, REM} group, which indicates that the hard/easy partition remains consistent across different age groups.

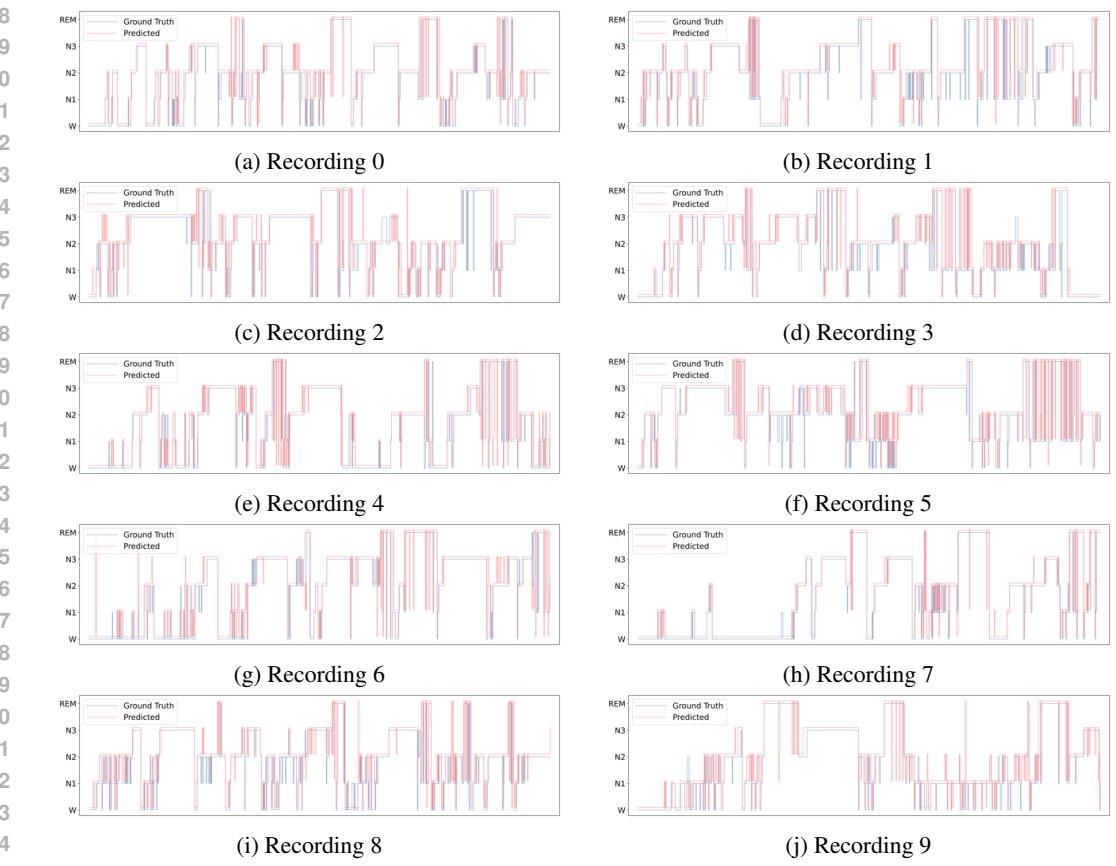

Figure 12: Sleep staging hypnogram comparisons on ISRUC-S3 across all 10 recordings (recording 0–recording 9).

## A.17 THEORETICAL ANALYSIS OF SAE EFFECTIVENESS

SAE architecture is based on a hierarchical, mixture-of-experts principle. Its operation consists of two distinct stages: first, a coarse-grained routing mechanism assigns an input sample to a high-level expert group; second, a fine-grained classification is performed within that selected group to yield the final prediction. Motivated by medical domain knowledge, we partition the five sleep stages into two groups: $G_1 = \{W, N1, N2\}$ and $G_2 = \{N3, REM\}$. This partitioning is empirically justified by the performance of single-stage five-class baselines (e.g., MVF-SleepNet and cVAN), which, while effective at separating $G_1$ from $G_2$, frequently confuse stages within $G_1$ (i.e., W, N1, and N2). By leveraging this discovery, the SAE effectively decomposes the complex, single-stage five-class problem into a simpler framework. It replaces one highly complex decision boundary with a single coarse boundary ($G_1$ vs. $G_2$) and two simpler, more localized sub-boundaries within each group. This hierarchical decomposition significantly reduces the geometric complexity that each expert model must learn, leading to more robust and accurate stage classification.

Formally, let $Y \in \{W, N1, N2, N3, REM\}$ be the ground-truth sleep stage label, and let $G_Y \in \{G_1, G_2\}$ be its corresponding group label. The SAE architecture consists of a coarse-grained grouping classifier that predicts a group $\hat{G}_Y$ and a set of group-specific expert classifiers that produce the final prediction $\hat{Y}$. The overall error rate of the SAE, $E_{SAE}$, can be decomposed as follows:

$$E_{SAE} = Pr(\hat{G}(Y) \neq G_Y) + Pr(\hat{G}(Y) = G_Y, \hat{Y} \neq Y)$$

Let $A_g$ denote the accuracy of the grouping classifier. Furthermore, for each group $g \in 1, 2$, let $E_g$ be the error rate of the corresponding expert, and let $\pi_g$ be the prior probability (proportion) of

group $g$. The total error rate can be reformulated as:

$$E_{SAE} = (1 - A_g) + A_g \sum_{g=1}^{2} \pi_g E_g$$

To compare the SAE architecture against a standard single-stage classifier, we define the error rate of the monolithic five-class classifier as $E_s$. For simplicity, let $\bar{E} = \sum_{g=1}^{2} \pi_g E_g$ represent the average within-group error rate of the experts, weighted by the group proportions.

Our objective is to establish the condition under which the SAE model outperforms the single classifier, *i.e.*, $E_{SAE} < E_s$. Substituting the expressions above, this inequality is equivalent to:

$$(1 - A_g) + A_g \bar{E} < E_s$$

Rearranging terms yields the following necessary and sufficient condition:

$$A_g(1 - \bar{E}) > 1 - E_s \tag{8}$$

This inequality holds under two intuitive and reasonable conditions:

1. High Grouping Accuracy: The grouping classifier is highly accurate, *i.e.*, $A_g > 1 - \epsilon$ for a small $\epsilon > 0$.

2. Superior Expert Performance: The experts, on average, achieve a lower error rate than the single classifier, *i.e.*, $\bar{E} \leq E_s - \delta$ for some $\delta > 0$.

**Proof**: From Condition 2, we have $1 - \bar{E} \geq 1 - E_s + \delta$. Combining this with Condition 1 ($A_g > 1 - \epsilon$), the left-hand side of inequality (8) satisfies:

$$A_g(1 - \bar{E}) > (1 - \epsilon)(1 - E_s + \delta)$$

For inequality (1) to hold, it is sufficient that:

$$A_g(1 - \bar{E}) > (1 - \epsilon)(1 - E_s + \delta)$$

Expanding and simplifying the left side:

$$(1 - E_s + \delta) - \epsilon(1 - E_s + \delta) > 1 - E_s$$

Subtracting $1 - E_s$ from both sides gives:

$$\delta = \epsilon(1 - E_s + \delta) > 0$$

This simplifies to the final sufficient condition:

$$\epsilon < \frac{\delta}{1 - E_s + \delta} \tag{9}$$

We verify that our empirical results on the ISRUC-S3 dataset satisfy these conditions. The grouping classifier achieves an accuracy of $A_g = 97.1\%$, implying $\epsilon = 0.029$. The The single-stage classifier (S$^3$Net without SAE) has an error rate of $E_s = 1 - 0.818 = 0.182$. The group-specific experts achieve error rates of $E_1 = 0.11$ and $E_2 = 0$, with group proportions $\pi_1 = 0.673$ and $\pi_2 = 0.327$. Thus, the average within-group error is:

$$\bar{E} = (0.673 \times 0.11) + (0.327 \times 0) = 0.074$$

The performance gain of the experts is $\delta = E_s - \bar{E} = 0.182 - 0.074 = 0.108$.

Substituting these values into condition (9):

$$\frac{\delta}{1 - E_s + \delta} = \frac{0.108}{1 - 0.182 + 0.108} = \frac{0.108}{0.926} \approx 0.1166$$

Since $\epsilon = 0.029 < 0.1166$, the sufficient condition (9) is satisfied. Therefore, under the realistic conditions of a highly accurate grouping classifier ($\epsilon$ is small) and experts that significantly reduce the within-group error ($\delta$ is sufficiently large), the theoretical inequality $E_{SAE} < E_s$ is guaranteed to hold, which is consistent with our empirical findings.

## A.18 THEORETICAL ANALYSIS OF T-ALN EXPRESSIVENESS

The t-ALN module serves as a bridge between the spectral and temporal branches, transforming spectral energy maps into a strict temporally organized query representation. A critical component of this transformation is the incorporation of stepwise positional encodings prior to flattening. This design preserves the temporal identity of features from the same time step, which is fundamental for capturing the sequential dynamics of sleep stages. We theoretically validate that incorporating positional information strictly enhances the model's expressive power compared to a position-agnostic baseline.

Formally, let $\mathcal{F}_\emptyset$ be a set of functions computable by a Transformer-based t-ALN module without positional encodings, and $\mathcal{F}_{\text{pos}}$ a set with stepwise positional encodings. To demonstrate a strict increase in expressive power, we establish the set inclusion:

$$\mathcal{F}_\emptyset \subset \mathcal{F}_{\text{pos}}.$$

This requires proving two properties:

- **Compatibility**: $\mathcal{F}_\emptyset \subseteq \mathcal{F}_{\text{pos}}$. The position-aware model can replicate all behaviors of the position-agnostic model.
- **Strict Inequality**: There must exist at least one function $f$ representing a valid temporal pattern such that $f \in \mathcal{F}_{pos}$ but $f \notin \mathcal{F}_\emptyset$.

**Proof of Compatibility**: For any function $F \in \mathcal{F}_\emptyset$, an equivalent function in $\mathcal{F}_{\text{pos}}$ can be constructed by setting all positional encoding vectors $p_i$ to zero. In this case, the input to the self-attention mechanism is $x_i + p_i = x_i$, rendering the position-aware model mathematically identical to its position-agnostic counterpart. Thus, $\mathcal{F}_\emptyset \subseteq \mathcal{F}_{\text{pos}}$ holds.

**Proof of Strict Dominance**: We demonstrate this by constructing a function that requires sensitivity to absolute positional order. Consider a sequence $S = [t_1, t_2, t_3, t_4, \dots]$ and the "Odd-Even Swap" function:

$$f_{\text{swap}}(S) = [t_2, t_1, t_4, t_3, \dots].$$

Models in $\mathcal{F}_\emptyset$ are permutation equivariant. For any permutation $\pi$ of the input indices, the output satisfies $F_\emptyset(\pi(S)) = \pi(F_\emptyset(S))$. However, $f_{\text{swap}}$ is not permutation equivariant. For example, applying a permutation $\pi$ that swaps the second and third elements yields:

$$f_{\text{swap}}(\pi(S)) \neq \pi(f_{\text{swap}}(S)).$$

Therefore, no function in $\mathcal{F}_\emptyset$ can implement $f_{\text{swap}}$. This validates our t-ALN design, proving that $\mathcal{F}_{\text{pos}}$ possesses a strictly superior theoretical capacity for modeling the temporal structure of sleep data.

