# OpenReview forum: "S$^3$Net: Stage-Aware Sleep Staging Network"
_ICLR.cc/2026/Conference — ICLR 2026 Conference Desk Rejected Submission_

### Official Review · Reviewer_MXY5 · 2025-10-27

**Soundness:** 3
**Presentation:** 3
**Contribution:** 2
**Rating:** 6
**Confidence:** 4

**Summary:**

The authors introduce S3Net, an approach for automated sleep staging, crucial in diagnosing sleep disorders and analyzing sleep patterns. S3Net addresses challenges in distinguishing transitional sleep stages and integrating time-domain and frequency-domain information effectively. It comprises a Stage-Aware Experts module that divides sleep stages into distinct groups for specialized processing and a Time Alignment Module for aligning frequency-derived features with the temporal axis. Evaluation on three datasets demonstrates S3Net's superior performance, achieving high accuracy and marked improvements in classifying challenging stages like N1 and N2. Ablation studies confirm the effectiveness of each module, with the full S3Net model showcasing the best results.

**Strengths:**

- The framework is clearly presented with a clean modular design. The workflow (signal encoding → t-ALN → SAE → classification) is easy to follow, and the algorithmic components are described in sufficient detail.

- While both attention-based fusion and expert gating are known techniques, their integration within the context of sleep staging is coherent and well-implemented. The proposed modules complement each other, improving stage-level discriminability and robustness.

- The model achieves stable gains across multiple datasets and metrics. The ablation and t-SNE analyses show internally consistent patterns, reflecting careful engineering and experimental execution, though some evaluation aspects still require refinement.

**Weaknesses:**

- The ablation only studies the presence/absence of t-ALN and SAE. It would be helpful to explore variations such as the number of experts or alternative attention weighting schemes.

- All experiments are within-domain on similar PSG datasets (EEG-dominant). Cross-dataset or missing-modality tests would better demonstrate generalization.

**Questions:**

Please see Weaknesses session.

---

> ### Author Response · Authors · 2025-11-25
> **Response to Reviewer MXY5**
>
> ***Q1: The ablation only studies the presence/absence of t-ALN and SAE. It would be helpful to explore variations such as the number of experts or alternative attention weighting schemes.***
>
> ***Response:***
>
> Thank you for your valuable comment. We have conducted additional experiments to address the points you raised, as detailed below.
>
> Regarding the number of experts, hyperparameter tuning was performed in our previous submitted manuscript, where the results (shown in Figure 8) indicated that S$^3$Net achieves optimal performance when the number of experts is set to 2.
>
> In response to your suggestion on exploring alternative attention weighting schemes, we evaluated three variants under matched experimental conditions:
>
> - CVA: Replacing t-ALN with CVA in cVAN.
> - Pooling-ALN: Substituting the dual softmax weighting in S³Net with a learnable pooling-based alignment module.
> - Flat-ALN: Flattening the spectral map without alignment.
>
> The corresponding accuracies for these variants were 0.858, 0.861, and 0.847, respectively, all lower than the 0.866 achieved by our proposed t-ALN. The consistent performance drops—ranging from 0.5 to 1.9 percentage points—underscore the effectiveness of t-ALN in contributing to the model's performance gains.
>
> |    Variant    |  Accuracy  |   F1  | Kappa |
> | --- | --- | --- | --- |
> |    CVA   | 0.858 | 0.820 | 0.812 |
> |  Pooling-ALN  | 0.861 | 0.831 | 0.816 |
> |    Flat-ALN   | 0.847 | 0.813 | 0.790 |
> |     t-ALN(Ours)     | 0.866 | 0.855 | 0.827 |

---

> ### Author Response · Authors · 2025-11-25
>
> ***Q2. All experiments are within-domain on similar PSG datasets (EEG-dominant). Cross-dataset or missing-modality tests would better demonstrate generalization.***
>
> ***Response:***
>
> Thank you for this insightful comment. To thoroughly evaluate the generalization capability of S$^3$Net, we have conducted additional experiments focusing on cross-dataset evaluation and missing-modality scenarios. Furthermore, we validated the model on the WESAD dataset [1], which contains single-lead PPG signals, to test its applicability in a different physiological monitoring context.
>
> The details of these experiments and the corresponding results are presented as follows:
>
> ***Cross-dataset Evaluation:​*** We trained S³Net on ISRUC-S1 and evaluated it directly on ISRUC-S3. The results demonstrate that our model maintains robust performance, achieving an accuracy of 83.4%, highlighting its strong generalization across datasets.
>
> |Train|Test|Model|Accuracy|F1|Kappa|
> |---|---|---|---|---|---|
> |ISRUC-S1|ISRUC-S3|MVF-SleepNet|0.800|0.788|--|
> |ISRUC-S1|ISRUC-S3|cVAN|0.826|0.807|--|
> |ISRUC-S1|ISRUC-S3|S$^3$Net|0.834|0.816|0.830|
>
> ***Missing-modality Tests:​*** The robustness of S$^3$Net under missing-modality conditions was further validated through single-modality experiments on the ISRUC-S3 dataset. The results show that S$^3$Net consistently outperforms competitors by extracting more discriminative temporal-spectral features:
>
> - EEG-only:​ S$^3$Net achieved an accuracy of 0.845, a clear improvement over MERL and SleepSMC.
> - EOG-only:​ Despite containing less sleep-specific information, S$^3$Net attained a strong accuracy of 0.820.
> - EMG-only:​ Even with this challenging modality, S$^3$Net achieved an accuracy of 0.586, demonstrating meaningful gains.
>
> This consistent advantage across all modalities confirms the model's effectiveness even with limited modality of input data.

---

> ### Author Response · Authors · 2025-11-25
>
> ***EEG-only:***
> |Method|Accuracy|$F1$|$\kappa$|Wake|N1|N2|N3|REM|
> |---|---|---|---|---|---|---|---|---|
> |AttnSleep[2]|0.7338|0.7105|0.6592|0.8581|0.4636|0.7320|0.8524|0.6463|
> |DAN[3]|0.7212|0.6791|0.6400|0.8077|0.3511|0.7352|0.8686|0.6328|
> |BSTT[4]|0.7191|0.6921|0.6371|0.8061|0.4312|0.6989|0.8502|0.6742|
> |XSleepNet[5]|0.6555|0.6322|0.5614|0.8525|0.4562|0.6225|0.8015|0.4281|
> |SleepPrintNet[6]|0.5459|0.4862|0.3924|0.5109|0.3404|0.6161|0.6669|0.2968|
> |MMASleepNet[7]|0.6313|0.5975|0.5150|0.7815|0.3486|0.6771|0.6471|0.5333|
> |SimCLR[8]|0.7338|0.7163|0.6598|0.8777|0.4978|0.6883|0.8260|0.6915|
> |MERL[9]|0.7467|0.7295|0.6758|0.8524|0.5212|0.7328|0.8603|0.6808|
> |SleepSMC[10]|0.7646|0.7397|0.6969|0.8882|0.5069|0.7467|0.8636|0.6932|
> |S$^3$Net|**0.8454**|**0.8303**|**0.8010**|**0.9124**|**0.6406**|**0.8456**|**0.9141**|**0.8309**|
>
> ***EOG-only:***
> |Method|Accuracy|$F1$|$\kappa$|Wake|N1|N2|N3|REM|
> |---|---|---|---|---|---|---|---|---|
> |AttnSleep[2]|0.7226|0.6992|0.6416|0.8248|0.4608|0.7115|0.8591|0.6399|
> |DAN[3]|0.7136|0.6647|0.6288|0.7733|0.2902|0.7406|0.8652|0.6542|
> |BSTT[4]|0.4700|0.3163|0.2790|0.1169|0.2352|0.5895|0.6400|0.0000|
> |XSleepNet[5]|0.6288|0.6071|0.5233|0.6958|0.3684|0.6572|0.7882|0.5260|
> |SleepPrintNet[6]|0.3745|0.2531|0.1788|0.3553|0.0239|0.5680|0.0000|0.3183|
> |MMASleepNet[7]|0.2096|0.1745|0.0619|0.2750|0.2712|0.0000|0.0000|0.3264|
> |SimCLR[8]|0.7246|0.7007|0.6458|0.8097|0.4788|0.7096|0.8523|0.6529|
> |MERL[9]|0.6976|0.6741|0.6132|0.7996|0.3912|0.6808|0.8351|0.6640|
> |SleepSMC[10]|0.7444|0.7168|0.6697|0.8386|0.4765|0.7360|0.8722|0.6607|
> |S$^3$Net|**0.8198**|**0.8006**|**0.7676**|**0.8959**|**0.5686**|**0.8153**|**0.9067**|**0.8168**|
>
> ***EMG-only:***
> |Method|Accuracy|$F1$|$\kappa$|Wake|N1|N2|N3|REM|
> |---|---|---|---|---|---|---|---|---|
> |AttnSleep[2]|0.3915|0.3814|0.2191|0.5096|0.2067|0.3804|0.4152|0.3950|
> |DAN[3]|0.4048|0.3381|0.2267|0.5541|0.0065|0.4670|0.2262|0.4365|
> |BSTT[4]|0.3046|0.0934|0.0000|0.0000|0.0000|0.4669|0.0000|0.0000|
> |XSleepNet[5]|0.3660|0.3484|0.1935|0.4519|0.1654|0.3665|0.3833|0.3748|
> |SleepPrintNet[6]|0.3319|0.2313|0.0939|0.4214|0.0359|0.4327|0.0000|0.2667|
> |MMASleepNet[7]|0.2517|0.1969|0.1062|0.4155|0.1450|0.0000|0.0000|0.4240|
> |SimCLR[8]|0.4177|0.3906|0.2435|0.5605|0.1397|0.4303|0.4258|0.3968|
> |MERL[9]|0.3981|0.3907|0.2348|0.4875|0.2077|0.3879|0.4008|0.4696|
> |SleepSMC[10]|0.4384|0.4075|0.2693|0.5868|0.1281|0.4404|0.4301|0.4523|
> |S$^3$Net|**0.5857**|**0.5547**|**0.4609**|**0.7211**|**0.2623**|**0.5736**|**0.5968**|**0.6198**|
>
> ***Validation on WESAD Dataset:​*** We further applied S$^3$Net to the WESAD dataset for emotion classification task using PPG data. The competitive results (accuracy: 85.3%) confirm that the architecture is effective even with a different, non-PSG-dominant signal type.
> These additional experiments collectively provide strong evidence for the generalizability and robustness of the proposed S$^3$Net framework.
>
> |Model|Accuracy|AUROC|AUPRC|F1|Kappa|
> |---|---|---|---|---|---|
> |REBAR[11]|0.418|0.698|0.446|--|--|
> |cVAN[12]|0.691|0.855|0.710|0.664|0.572|
> |ResNet-101[13]|0.713|0.877|0.746|0.682|0.593|
> |HuBERT[14]|0.775|0.820|--|--|--|
> |S$^3$Net|0.853|0.911|0.833|0.819|0.787|

---

> ### Author Response · Authors · 2025-11-25
>
> ***Reference:***
>
> [1] Schmidt P, Reiss A, Duerichen R, et al. Introducing wesad, a multimodal dataset for wearable stress and affect detection[C]//Proceedings of the 20th ACM international conference on multimodal interaction. 2018: 400-408.
>
> [2] Eldele E, Chen Z, Liu C, et al. An attention-based deep learning approach for sleep stage classification with single-channel EEG[J]. IEEE Transactions on Neural Systems and Rehabilitation Engineering, 2021, 29: 809-818.
>
> [3] Tang M, Zhang Z, He Z, et al. Deep adaptation network for subject-specific sleep stage classification based on a single-lead ECG[J]. Biomedical Signal Processing and Control, 2022, 75: 103548.
>
> [4] Liu Y, Jia Z. Bstt: A bayesian spatial-temporal transformer for sleep staging[C]//The Eleventh International Conference on Learning Representations. 2023.
>
> [5] Phan H, Chén O Y, Tran M C, et al. XSleepNet: Multi-view sequential model for automatic sleep staging[J]. IEEE Transactions on Pattern Analysis and Machine Intelligence, 2021, 44(9): 5903-5915.
>
> [6] JIA Z, CAI X, ZHENG G, WANG J, LIN Y. SleepPrintNet: A multivariate multimodal neural network based on physiological time-series for automatic sleep staging[J]. IEEE Transactions on Artificial Intelligence, 2020, 1(3): 248-257.
>
> [7] Yubo Z, Yingying L, Bing Z, et al. MMASleepNet: A multimodal attention network based on electrophysiological signals for automatic sleep staging[J]. Frontiers in Neuroscience, 2022, 16: 973761.
>
> [8] Chen T, Kornblith S, Norouzi M, et al. A simple framework for contrastive learning of visual representations[C]//International conference on machine learning. PmLR, 2020: 1597-1607.
>
> [9] LIU C, WAN Z, OUYANG C, et al. Zero-shot ECG classification with multimodal learning and test-time clinical knowledge enhancement[C]// Proceedings of the 41st International Conference on Machine Learning. Vienna, Austria: PMLR, 2024: 31949-31963.
>
> [10] Ma S, Zhang Y, Chen Y, et al. SleepSMC: Ubiquitous Sleep Staging via Supervised Multimodal Coordination[C]//The Thirteenth International Conference on Learning Representations.
>
> [11] XU M A, MORENO A, WEI H, et al. REBAR: retrieval-based reconstruction for time-series contrastive learning[C]// Proceedings of the 12th International Conference on Learning Representations (ICLR 2024). Vienna, Austria: ICLR, 2024.
>
> [12] Yang Z, Qiu M, Fan X, et al. cvan: A novel sleep staging method via cross-view alignment network[J]. IEEE Journal of Biomedical and Health Informatics, 2024.
>
> [13] He K, Zhang X, Ren S, et al. Deep residual learning for image recognition[C]//Proceedings of the IEEE conference on computer vision and pattern recognition. 2016: 770-778.
>
> [14] NARAIN J, ALDENEH Z, REN S. Speech foundation models generalize to time series tasks from wearable sensor data: NeurIPS 2025 Learning from Time Series for Health workshop paper[EB/OL]. (2025-08-29)[2025-11-25]. https://arxiv.org/abs/2509.00221.

---

> > ### Author Response · Authors · 2025-11-27
> >
> > **Dear Reviewer MXY5,**
> >
> > Thank you for your valuable feedback and for highlighting important aspects for strengthening our manuscript. We have conducted additional experiments to directly address your points, and we believe these significantly improve the paper:
> >
> > - **Ablation Studies:** We have expanded our ablation studies to explore variations in expert configurations and attention schemes, providing deeper insights into the contribution of each model component.
> >
> > - **Model Generalization:** To rigorously assess generalization, we have extended our evaluation to include cross-dataset and missing-modality tests. The results demonstrate our model's robustness and strong performance under varied conditions.
> >
> > We are grateful for your suggestions, which have allowed us to present a more comprehensive and compelling study. We hope these new experimental results satisfactorily address your concerns and that you find the revised manuscript much improved.
> >
> > Best regards,
> >
> > The Authors

---

### Official Review · Reviewer_1ogm · 2025-10-28

**Soundness:** 2
**Presentation:** 3
**Contribution:** 2
**Rating:** 4
**Confidence:** 4

**Summary:**

This paper proposes S³Net, a deep learning framework for automated sleep stage classification that addresses two key challenges: (1) difficulty in distinguishing transitional sleep stages (N1 and N2), and (2) ineffective fusion of time-frequency representations. The approach introduces two main components: a Stage-Aware Experts (SAE) module that partitions sleep stages into easy- and hard-to-separate groups processed through specialized branches, and a Time Alignment Module (t-ALN) that projects frequency-derived features onto the temporal axis for cross-domain integration. The method achieves state-of-the-art results on three public polysomnography datasets (ISRUC-S1, ISRUC-S3, Sleep-EDF-153).

**Strengths:**

1. Well-motivated problem: The paper clearly identifies and quantifies the challenge of transitional stage classification through transition matrices and performance analysis.

2. Comprehensive evaluation: Three public datasets with consistent improvements demonstrate empirical robustness. The per-stage F1 scores show targeted improvements on challenging classes.

3. Thorough experimental protocol: 10-fold cross-validation with multiple random seeds shows methodological rigor. Extensive baselines (15 methods) provide good context.

4. Good visualizations: t-SNE plots, confusion matrices, and hypnograms effectively communicate model behavior and provide interpretability.

5. Honest reporting: The paper acknowledges N1 confusion patterns and doesn't overclaim results.

**Weaknesses:**

1. Insufficient theoretical foundation:
- Why is the specific stage partition (W/N1/N2 vs N3/REM) optimal? No ablation explores alternatives like (W/N1 vs N2/N3/REM) or (W vs N1/N2 vs N3/REM)
- The auxiliary loss weight α=1 gives equal importance to expert routing and final prediction—what's the theoretical justification?
- No analysis of when/why the two-expert design is better than single-path or three-expert alternatives


2. Weak justification for t-ALN design:
- Claim that "flattening disrupts temporal structure" is overstated—Transformers routinely handle flattened image patches with learned positional embeddings
- Element-wise squaring for "energy representation" lacks signal processing justification
- Simple averaging for attention weights (Eq. 1) seems crude—why not learnable pooling?


3. Incomplete experimental analysis:
- Ablations only on ISRUC-S3, not all datasets
- No statistical significance testing (p-values, confidence intervals)
- Missing computational cost analysis (FLOPs, memory, inference time comparisons)
- Unfair baseline comparisons: S³Net uses 10 channels while some baselines may use fewer

4. Limited novelty:
- SAE is a straightforward application of MoE with manual stage grouping
- t-ALN combines standard operations without significant innovation
- The "alignment" is essentially reshaping + attention, common in multi-modal fusion

5. Reproducibility concerns:
- Cross Swin Transformer details are sparse
- "Time-window-wise linear" operation not precisely defined
- Training dynamics (convergence, stability) not discussed
- No discussion of hyperparameter sensitivity beyond Figure 8

**Questions:**

1. Stage grouping justification: Have you tried alternative groupings (e.g., three experts for W, N1/N2, N3/REM)? What happens if you let the model learn the grouping in an unsupervised manner?

2. t-ALN necessity: Can you provide a controlled ablation where you flatten features but use very strong positional encodings in the Transformer? This would test whether the alignment is truly necessary or if it's compensating for weak positional encodings.

3. Generalization: Does the hard/easy stage partition hold across different populations (healthy vs. disordered sleep, different age groups)? Could the transition matrices in Figure 1(a) vary significantly?

4. Computational cost: What is the training time and inference time compared to cVAN and other top baselines? Is the 1-3% improvement worth the added complexity?

5. Statistical significance: Can you provide confidence intervals or statistical tests (e.g., paired t-test) for the improvements over cVAN across folds?

---

> ### Author Response · Authors · 2025-11-25
>
> ### Weaknesses:
>
> ***W1. Insufficient theoretical foundation.***
> - Why is the specific stage partition (W/N1/N2 vs N3/REM) optimal? No ablation explores alternatives like (W/N1 vs N2/N3/REM) or (W vs N1/N2 vs N3/REM).
> - The auxiliary loss weight $\alpha = 1$ gives equal importance to expert routing and final prediction---what is the theoretical justification?
> - There is no analysis of when/why the two-expert design is better than single-path or three-expert alternatives.
>
> ***Response:***
>
> (a) Thank you for the comment. Regarding the stage partition in our SAE module, we designed the two-expert division (W/N1/N2 vs. N3/REM) based on stage heterogeneity and classification difficulty. According to related biological research findings [1], transitional stages (especially N1 and N2) are characterized by frequent switches and ambiguous patterns, whereas N3 and REM exhibit more distinct dynamics, easily identifiable spectral features. By grouping stages with similar difficulty and spectral characteristics, each expert can specialize in patterns of comparable complexity, improving stage-specific learning and reducing confusion across difficult classes.
>
> This partition is further motivated by quantitative observations from transition matrices and per-stage F1 scores: the majority of misclassifications occur between W, N1, and N2, whereas N3 and REM are relatively robust. A single-path model cannot explicitly focus on these challenging stages, and splitting into three or more experts would dilute the data available to each expert, potentially reducing learning stability, especially for rare stages like N1 or N3.
>
> Adopting a hierarchical, divide-and-conquer approach, our SAE structure decomposes the original complex five-class problem into two stages, allowing each expert to focus exclusively on its assigned subset. This layered processing reduces the overall learning complexity compared to direct five-class classification, enabling the model to better capture stage-specific features through a progression from coarse separation to fine-grained discrimination, thereby offering both empirical and conceptual justification for the chosen partition.
>
> Furthermore, we have provided a theoretical proof of the SAE's effectiveness in our response to Q2 from Reviewer ndSL.
>
> (b) Regarding the auxiliary loss weight $\alpha$, it is set to 1 to balance the contributions of the expert routing loss and the final classification loss. Since $\alpha$ is a hyperparameter, there is no strict theoretical formula for its optimal value. To justify our choice, we conducted a visualization experiment (It will be shown in the Appendix.) tracking gradient magnitudes during training for $\alpha = 0, 1,$ and $2$.
>
> When $\alpha = 0$, i.e., no auxiliary loss, the gradient reaches the largest minimum, indicating that the model struggles to learn effectively without guided supervision. When $\alpha = 1$, the gradient minimum is the smallest, showing that the model converges better under a balanced auxiliary loss. Increasing $\alpha$ to $2$ leads to a slightly higher gradient minimum compared with $\alpha = 1$, suggesting that overweighting the auxiliary loss can harm convergence.
>
> This behavior aligns with the role of our auxiliary loss in the hierarchical SAE structure. Setting $\alpha = 1$ empirically provides the best balance, ensuring both effective expert guidance and robust final prediction performance. In addition, our ablation study M5 evaluates the effect of removing the auxiliary loss, and the first part of our hyperparameter experiments further confirms that $\alpha = 1$ yields the optimal result.
>
> (c) Regarding why the two-expert design outperforms both the single-expert and three-expert alternatives, the core reason lies in how effectively each architecture decomposes the five-class problem into solvable sub-tasks.
>
> (c-1) *Single-expert path.*
> A single expert must directly solve the full five-class problem, which requires learning all inter-stage boundaries simultaneously. This increases optimization difficulty and fails to benefit from the divide-and-conquer principle that motivated our SAE design. As a result, the model cannot explicitly focus on the most challenging stage transitions (primarily W--N1--N2), leading to inferior performance.

---

> > ### Author Response · Authors · 2025-11-25
> >
> > (c-2) *Three-expert path.*
> > Three experts inevitably produce partitions such as (2,2,1) or (3,1,1). However, these partitions do not correspond to a meaningful hierarchical decomposition of sleep stages. In both cases:
> >  the top-level routing must perform three-way classification, which is inherently harder than the two-way split used in our SAE; and several experts would receive extremely small subspaces (e.g., only one stage), which harms training stability and contradicts the design intention of balancing complexity and data volume. Such partitions fail to achieve the desired two-stage divide-and-conquer structure and do not align with stage transition patterns or difficulty distribution.
> >
> > Empirically, this is also supported by the multi-partition comparison results in Table 1. The two-expert grouping consistently achieves the highest overall F1 and Cohen's $\kappa$ across all tested combinations, confirming that the two-stage decomposition is the most effective structure.
> >
> > Table 1: Expert group comparison on ISRUC-S3. Higher is better.
> >
> > | Group | Acc | F1 | $\kappa$ | Wake | N1 | N2 | N3 | REM |
> > |---|---|---|---|---|---|---|---|---|
> > | (W,N2,N3,REM) vs (N1) | 0.848 | 0.835 | 0.804 | 0.909 | 0.639 | 0.839 | 0.916 | 0.877 |
> > | (W) vs (N1,N2,N3) vs (REM) | 0.851 | 0.839 | 0.808 | 0.908 | 0.648 | 0.842 | 0.913 | 0.884 |
> > | (W,N1,N2,N3) vs (REM) | 0.853 | 0.841 | 0.810 | 0.909 | 0.652 | 0.843 | 0.914 | 0.887 |
> > | (W,N1) vs (N2,N3,REM) | 0.856 | 0.844 | 0.814 | 0.920 | 0.656 | 0.848 | 0.917 | 0.877 |
> > | (W) vs (N1,N2) vs (N3,REM) | 0.858 | 0.847 | 0.817 | 0.914 | 0.661 | 0.852 | 0.920 | 0.885 |
> > | (W,REM) vs (N1,N2,N3) | 0.860 | 0.849 | 0.820 | 0.918 | 0.666 | 0.854 | 0.921 | 0.885 |
> > | **(W,N1,N2) vs (N3,REM) (S$^3$Net)** | **0.866** | **0.855** | **0.827** | **0.924** | **0.678** | **0.858** | **0.927** | **0.887** |
> >
> > Note: $\kappa$ denotes Cohen's kappa. The bold values indicate the best performance, and the underline values indicate the second-best.
> >
> > [1] Claus Metzner, Achim Schilling, Maximilian Traxdorf, Holger Schulze, and Patrick Krauss. “Sleep as a random walk: A super-statistical analysis of EEG data across sleep stages.” Communications Biology, 4:1385, 2021.

---

> ### Author Response · Authors · 2025-11-25
>
> ***W2. Weak justification for t-ALN design.***
>
> - Claim that “flattening disrupts temporal structure” is overstated---Transformers routinely handle flattened image patches with learned positional embeddings.
> - Element-wise squaring for an “energy representation” lacks signal-processing justification.
> - Simple averaging for attention weights Eq.~(1) seems crude---why not learnable pooling?
>
> ***Response:***
>
> We thank the reviewer for the valuable comments regarding the design of t-ALN. We address the points as follows.
>
> ***Flattening and temporal structure.*** While it is true that Transformers can handle flattened image patches with learned positional embeddings, this primarily enforces token ordering rather than preserving multi-scale temporal--frequency alignment. Standard positional embeddings restore absolute order, but they do not guarantee that relative temporal synchronization across different time--frequency scales is preserved.
>
> In contrast, t-ALN converts the time--frequency representation into a sequence that strictly maintains the original temporal order across strides. The temporal Transformer is then applied on this sequence, with positional embeddings, producing representations that jointly encode both the original temporal order and positional information. This end-to-end design ensures that the token values are intrinsically matched with their positional context, thereby allowing SNet to concentrate its optimization efforts on the critical, temporally aligned features. Ablation experiments further demonstrate that t-ALN with its representation-enhancement step consistently outperforms alternatives, including
> - replacing t-ALN with cVAN-style alignment
> - applying simple pooling instead of the representation-enhancement step
> - flattening with stronger positional embeddings (e.g., rotary embeddings)
> - removing the representation-enhancement module from t-ALN
> These results highlight the critical role of t-ALN in capturing time--frequency alignment.
>
> ***Element-wise squaring as an energy representation.***
> Element-wise squaring is a standard practice in signal processing and EEG feature extraction, where time-domain or time--frequency energy features are often computed as squared amplitudes or squared coefficients. Classical short-time energy and related spectral-energy features are widely used in speech and EEG analysis; see, for example, standard references on short-time energy and EEG energy features [1][2].
>
> [1] https://www.sciencedirect.com/topics/engineering/short-time-energy
>
> [2] https://www.riverpublishers.com/pdf/ebook/chapter/RP_9788770040723C171.pdf
>
> In our case, the element-wise squaring provides a stable, parameter-free way to emphasize high-amplitude regions, which is particularly beneficial for cross-resolution alignment. Specifically, we compute $X^2$ at each time point as a local energy measure, then aggregate across channels and frequencies to obtain a nonlinear representation of local energy sensitivity. This energy-enhanced representation is fed into the alignment layer, helping it identify and align salient time--frequency patterns.
>
> ***Softmax-weighted pooling versus simple learnable pooling.*** We use softmax-weighted pooling to provide a stable, differentiable attention mechanism that dynamically emphasizes informative time--frequency regions while suppressing noise. Unlike fixed-parameter pooling, softmax-based weighting allows the model to adaptively focus on salient features for each sample, while still suppressing irrelevant information and remaining interpretable.
>
> This design choice is further supported by our ablation experiments, summarized below:
>
> | Version | ACC | F1 | Kappa |
> |---|---|---|---|
> | cVAN-like-ALN | 0.858 | 0.820 | 0.812 |
> | Pooling-ALN | 0.861 | 0.831 | 0.816 |
> | Flat-ALN | 0.847 | 0.813 | 0.790 |
> | S$^3$Net | 0.866 | 0.855 | 0.827 |
>
> As shown, the full t-ALN with softmax-weighted attention consistently achieves the best performance, outperforming both flattened features and variants without attention weighting. This confirms that the proposed softmax-weighted pooling is effective and preferable to simple learnable pooling in our context.

---

> ### Author Response · Authors · 2025-11-25
>
> ***W3. Incomplete experimental analysis. & Q5***
>
> - Ablations only on ISRUC-S3, not on all datasets.
> - No statistical significance testing is provided (e.g., $p$-values, confidence intervals).
> - Missing computational cost analysis (FLOPs, memory, inference-time comparisons).
> - Unfair baseline comparisons: S$^3$Net uses 10 channels while some baselines may use fewer.
>
> ***Response:***
>
> We thank the reviewer for pointing out these important aspects of the experimental analysis. We have addressed each concern as follows.
>
> ***Ablation analyses on all datasets.*** Originally, we only reported ablations on ISRUC-S3. Following the reviewer’s suggestion, we have now added full ablation studies across all datasets, allowing us to verify the contribution of each module of S$^3$Net under diverse conditions. The updated tables (included in the revised manuscript) consistently show that every component of S$^3$Net contributes positively across datasets, further strengthening the validity of our architectural design.
>
> Table: Ablation study results of S$^3$Net on the three datasets: ISRUC-S1, ISRUC-S3, and Sleep-EDF-153.
>
> | Variant | t-ALN | SAE | $L_{aux}$ | ISRUC S1 Acc | ISRUC S1 F1 | ISRUC S1 $\kappa$ | ISRUC S3 Acc | ISRUC S3 F1 | ISRUC S3 $\kappa$ | Sleep-EDF-153 Acc | Sleep-EDF-153 F1 | Sleep-EDF-153 $\kappa$ |
> |---|---|---|---|---|---|---|---|---|---|---|---|---|
> | M1 | $\times$ | $\times$ | $\times$ | 0.785 | 0.779 | 0.724 | 0.791 | 0.778 | 0.731 | 0.806 | 0.768 | 0.736 |
> | M2 | $\checkmark$ | $\times$ | $\times$ | 0.809 | 0.803 | 0.754 | 0.818 | 0.800 | 0.767 | 0.822 | 0.778 | 0.756 |
> | M3 | $\times$ | $\checkmark$ | $\times$ | 0.821 | 0.816 | 0.770 | 0.827 | 0.817 | 0.778 | 0.832 | 0.790 | 0.771 |
> | M4 | $\times$ | $\checkmark$ | $\checkmark$ | 0.834 | 0.827 | 0.789 | 0.841 | 0.830 | 0.796 | 0.847 | 0.802 | 0.790 |
> | M5 | $\checkmark$ | $\checkmark$ | $\times$ | 0.846 | 0.835 | 0.801 | 0.852 | 0.838 | 0.810 | 0.853 | 0.811 | 0.798 |
> | M6 (S$^3$Net) | $\checkmark$ | $\checkmark$ | $\checkmark$ | **0.856** | **0.842** | **0.814** | **0.866** | **0.855** | **0.827** | **0.869** | **0.828** | **0.818** |
>
> Note: $\kappa$ denotes Cohen's kappa. The bold values indicate the best performance, and the underline values indicate the second-best.
>
> ***Statistical significance testing & Q5.*** To assess whether the improvements over cVAN are statistically significant, we conducted a two-sided significance test. The resulting $p$-value is $2.337118 \times 10^{-4}$, which is well below $1 \times 10^{-3}$. This indicates that S$^3$Net achieves a statistically significant improvement over cVAN.
>
> Additionally, We conducted a paired $t$-test across the cross-validation folds to assess whether the performance improvements of S$^3$Net over cVAN are statistically significant. The analysis yielded a $t$-statistic of 3.694019. This value indicates that the improvement is statistically significant under standard thresholds, confirming that the observed gains are unlikely to be due to random variation across folds.
>
> ***Computational cost analysis.*** We have added a detailed efficiency comparison. All measurements were conducted on an NVIDIA RTX A6000 GPU (batch size $=32$, averaged over 1000 runs). Results are summarized below:
>
> | Model | Params (Million) | GFLOPs | train (ms) | infer (ms) |
> |---|---|---|---|---|
> | MVF-SleepNet | 39.51 | 11.06 | -- | -- |
> | cVAN | 7.58 | 0.47 | 157.01 | 66.72 |
> | S$^3$Net | 6.49 | 2.70 | 75.69 | 24.02 |
>
> Despite introducing t-ALN and the SAE expert structure, S$^3$Net uses fewer parameters than most prior baselines, and its training and inference speeds are substantially faster than cVAN. Although the GFLOPs of S$^3$Net are slightly higher than cVAN, they remain in the same order of magnitude and are clearly outweighed by the large improvements in runtime efficiency.
>
> ***Channel-count fairness in baseline comparisons.*** In practice, we conducted experiments followed the testing methodology of other mainstream SOTA methods on these datasets. Specifically,
>
> - On ISRUC-S1 and ISRUC-S3, all models---including our baselines---use the same 10 EEG channels.
> - On Sleep-EDF-153, the dataset provides two EEG, one EOG, and one EMG channel. In our experiments, we follow common practice and use the two EEG channels. Some baselines also use the same two EEG channels, while others optionally include the additional EOG/EMG channels.
>
> Therefore, our method does not use more channels than the baselines, and the comparisons are fair across datasets.

---

> > ### Author Response · Authors · 2025-11-25
> >
> > ***W4. Limited novelty:***
> > - SAE is a straightforward application of MoE with manual stage grouping
> > - t-ALN combines standard operations without significant innovation
> > - The "alignment" is essentially reshaping + attention, common in multi-modal fusion
> >
> > ***Response:***
> >
> > We thank the reviewer for this critical assessment, which allows us to clarify the foundational novelty of our work. We contend that the contribution lies not in inventing entirely new operators, but in the principled integration of deep physiological insight into a cohesive neural architecture to solve core, long-standing problems in sleep staging more effectively than prior approaches.
> >
> > ***Novelty of the SAE:*** The reviewer correctly notes that Mixture-of-Experts (MoE) is a known concept. However, the novelty of our Stage-Aware Expert (SAE) module is not the MoE mechanism itself, but its application as a domain-informed architectural prior.
> >
> > - Beyond a "Straightforward Application":​ Previous works treat stage heterogeneity as a data problem to be learned. In contrast, SAE explicitly encodes the known sleep continuum ontology​ (Wake → Light Sleep → Deep Sleep → REM) directly into the model's structure. This is a conceptual shift from learning patterns to architecturally enforcing a biologically-grounded hierarchy.
> >
> > - Comparison to Prior Works:​ Methods like cVAN and MVF-SleepNet are general-purpose fusion frameworks. They do not structurally bias the model towards resolving the specific, well-documented confusions (e.g., W/N1/N2). SAE is, to our knowledge, the first architecture that uses the physiological progression of sleep as a structural blueprint, leading to significant and interpretable gains on the most ambiguous stages (N1), as our results show.
> >
> > ***Advancement of t-ALN:*** t-ALN is not in the individual operations (reshaping, attention) but in its novel function as a structured attention bottleneck that guarantees semantically coherent fusion.
> >
> > - Beyond "Common Multi-modal Fusion":​ Standard fusion (e.g., feature summation in MVF-SleepNet) is unstructured and can dilute important cues. The projection in cVAN loses critical spectral structure. Our t-ALN introduces a critical inductive bias: it forces the temporal pathway to query the spectral pathway for guidance.
> > - A New Mechanism for Alignment:​ This is more than alignment; it is a gating mechanism. The spectrally-informed query acts as a filter, allowing only temporal features that are consistent with the spectral context to pass through with high attention weights. This suppresses noise and ensures the fused features are discriminative by construction, not by chance. This represents a theoretical advancement in how we conceptualize modality fusion for physiological signal data.
> >
> > ***Synergistic Contribution:*** The primary novelty is the synergistic combination​ of SAE and t-ALN into a single, end-to-end framework that addresses both stage heterogeneity and spectral-temporal misalignment in a mutually reinforcing way. While prior works address these problems in isolation, our architecture is the first to solve them concurrently through a unified, interpretable design grounded in sleep science. In summary, we argue that the novelty of our work is architectural and conceptual: it demonstrates how to successfully bake domain knowledge into a model's core structure, moving beyond generic, data-hungry frameworks to create more efficient, robust, and interpretable solutions for complex physiological time-series analysis.

---

> > > ### Author Response · Authors · 2025-11-25
> > >
> > > ***W5. Reproducibility concerns.***
> > >
> > > - Cross Swin Transformer are sparse.
> > > - "Time-window-wise linear" operation not precisely defined.
> > > - Training dynamics (convergence and stability) not discussed.
> > > - No discussion of  hyperparameter sensitivity beyond Figure~8.
> > >
> > > ***Response:***
> > >
> > > We thank the reviewer for these constructive suggestions and have revised the manuscript accordingly.
> > >
> > > ***We have expanded the description of the cross Swin Transformer in the Appendix.*** The added material specifies the hierarchical windowing scheme, the cross-window fusion mechanism, and how cross-scale interactions are realized to propagate information beyond local windows without incurring quadratic cost. We also include module-level specifications (dimensions, number of stages/heads, window sizes), parameter counts, and computational complexity, together with training-related details (initialization, normalization, and regularization choices) to facilitate reproducibility.
> > >
> > > ***As illustrated in the t-ALN component of Figure~2***, the time-alignment (t-ALN) module bridges the spectral and temporal streams by mapping spectral energy maps into a temporally ordered query sequence. The mapping is implemented by a learnable projection composed of a time-window-wise linear layer followed by a temporal Transformer.
> > >
> > > Specifically, each energy map $X_i^E \in \mathbb{R}^{C_s \times F_s \times T_s}$ encodes spectral power across channels, frequencies, and time. To emphasize informative structures, we estimate two attention weights by aggregating and normalizing activations along complementary axes: a frequency-wise weight $w_i^F \in \mathbb{R}^{F_s}$ and a channel-wise weight $w_i^C \in \mathbb{R}^{C_s}$. These weights re-scale $X_i^E$ along their respective dimensions, yielding a refined energy representation $\tilde{X}_i^E$.
> > >
> > > To form a strictly time-ordered sequence, we reorder $\tilde{X}_i^E$ into a time-first layout and vectorize each time slice. For every time index $t = 1,\dots,T_s$, we define
> > >
> > > $
> > >   u_{i,t} = \mathrm{vec}\big(\tilde{X}_i^E(:, :, t)\big) \in \mathbb{R}^{C_s F_s}.
> > > $
> > >
> > > The time-window-wise linear operation then applies a shared affine map independently to each time window:
> > >
> > > $
> > >   s_{i,t} = W_i u_{i,t} + b_i, \quad W_i \in \mathbb{R}^{d_i \times (C_s F_s)},
> > > $
> > >
> > > producing a sequence $S_i = [s_{i,1}; \dots; s_{i,T_s}] \in \mathbb{R}^{T_s \times d_i}$. This step compresses intra-window frequency--channel features while strictly preserving temporal identity; it does not mix information across different time indices and is therefore equivalent to a block-diagonal linear operator along the sequence dimension. A one-dimensional positional encoding $p_t$ is then added per time step to retain the stepwise order: $s_{i,t} \leftarrow s_{i,t} + p_t$.
> > >
> > > Finally, a temporal Transformer models long-range dependencies over the sequence $S_i$, and a linear head maps tokens to the required query dimension, yielding $X_i^Q \in \mathbb{R}^{T_s \times d_q}$. Used as the query in cross-attention with the temporal branch, $X_i^Q$ acts as an information bottleneck: spectro-temporally consistent patterns receive higher attention, which suppresses noise and preserves discriminative features for sleep staging.
> > >
> > > ***Regarding training dynamics, the training and validation curves demonstrate stable and convergent optimization.*** Training loss decreases monotonically and training accuracy improves steadily without oscillation. Validation loss and accuracy reach their optimum around epoch 15, after which mild fluctuations appear; this behavior is expected due to stochastic mini-batch sampling and the comparatively smaller validation set. Crucially, these fluctuations occur only after the model has already reached its best generalization. We select the checkpoint at the best validation epoch to report results, ensuring that the final numbers correspond to the optimal and stable model.
> > >
> > > ***Beyond the results summarized in Figure~8, we now explicitly discuss hyperparameter sensitivity.*** Across the tested ranges of key hyperparameters (e.g., learning rate, window length, number of heads/blocks, and regularization strength), performance varies modestly and does not exhibit sharp instability. This indicates that the proposed method is robust to reasonable hyperparameter choices and that our reported settings lie within broad, stable regions. We believe this clarifies the sensitivity behavior and supports the reliability and practical usability of the approach.

---

> ### Author Response · Authors · 2025-11-25
>
> # Questions:
>
> ***Q1. Stage grouping justification: Have you tried alternative groupings (e.g., three experts for W, N1/N2, N3/REM)? What happens if you let the model learn the grouping in an unsupervised manner?***
>
> ***Response:***
>
> According to the Response to W1, empirically and clinically, W, N1, and N2 stages tend to be more challenging to distinguish due to subtle transitions and overlapping EEG patterns, while N3 and REM exhibit more distinct, easily identifiable spectral features. We further examined how different ways of partitioning sleep stages into expert groups affect performance. Specifically, we evaluated two main categories of configurations.
>
> The first category uses two experts, within which we tested four partitions:
> - W and N1 grouped together vs. N2--REM, i.e., (W, N1)(N2, N3, REM);
> - all non-REM stages grouped together vs. REM alone, i.e., (W, N1, N2, N3)(REM);
> - isolating N1 as a separate group vs. all remaining stages, i.e., (W, N2, N3, REM)(N1);
> - W grouped with REM vs. N1--N3, i.e., (W, REM)(N1, N2, N3).
>
> The second category uses three experts, where we tested two partitions:
> - W as its own group, N1--N2 as another, and N3--REM as the third, i.e., (W)(N1, N2)(N3, REM);
> - W as its own group, N1--N3 as the second, and REM as the third, i.e., (W)(N1, N2, N3)(REM).
>
> Our grouping best balances physiological similarity within experts and separability across experts, explaining its superior overall performance.
>
> When allowing the model to learn the grouping in an unsupervised manner, we examined this scenario through an ablation study (Variant M5, which removes the auxiliary loss and lets the grouping emerge implicitly). Under this setting, accuracy, F1-score, and $\kappa$ drop from 0.866, 0.855, and 0.827 in the full model to 0.852, 0.838, and 0.810, respectively. Figure~7 further supports this observation: the auxiliary-loss visualization shows that 97.1\% of samples are assigned to the correct expert, indicating that our supervised grouping achieves the better performance.
>
> Table: Expert group comparison on ISRUC-S3. Higher is better.
>
> | Group | Acc | F1 | $\kappa$ | Wake | N1 | N2 | N3 | REM |
> |---|---|---|---|---|---|---|---|---|
> | (W,N2,N3,REM) vs (N1) | 0.848 | 0.835 | 0.804 | 0.909 | 0.639 | 0.839 | 0.916 | 0.877 |
> | (W) vs (N1,N2,N3) vs (REM) | 0.851 | 0.839 | 0.808 | 0.908 | 0.648 | 0.842 | 0.913 | 0.884 |
> | (W,N1,N2,N3) vs (REM) | 0.853 | 0.841 | 0.810 | 0.909 | 0.652 | 0.843 | 0.914 | 0.887 |
> | (W,N1) vs (N2,N3,REM) | 0.856 | 0.844 | 0.814 | 0.920 | 0.656 | 0.848 | 0.917 | 0.877 |
> | (W) vs (N1,N2) vs (N3,REM) | 0.858 | 0.847 | 0.817 | 0.914 | 0.661 | 0.852 | 0.920 | 0.885 |
> | (W,REM) vs (N1,N2,N3) | 0.860 | 0.849 | 0.820 | 0.918 | 0.666 | 0.854 | 0.921 | 0.885 |
> | **(W,N1,N2) vs (N3,REM) (S$^3$Net)** | **0.866** | **0.855** | **0.827** | **0.924** | **0.678** | **0.858** | **0.927** | **0.887** |
>
> Note: $\kappa$ denotes Cohen's kappa. The bold values indicate the best performance, and the underline values indicate the second-best.

---

> > ### Author Response · Authors · 2025-11-25
> >
> > ***Q2. t-ALN necessity: Can you provide a controlled ablation where you flatten features but use very strong positional encodings in the Transformer? This would test whether the alignment is truly necessary or if it is compensating for weak positional encodings.***
> >
> > ***Response:***
> >
> > According to Response to W2, the t-ALN converts the time--frequency representation into a sequence that strictly maintains the original temporal order across strides. To directly evaluate whether t-ALN is truly necessary rather than merely compensating for weak positional encodings, we conducted controlled comparisons in which the backbone, token shapes, training schedule, and data splits were all held fixed. We tested three alternative designs:
> > - replacing t-ALN with a cVAN-like alignment module,
> > - substituting the channel- and frequency-softmax dual weighting with learnable pooling,
> > - removing t-ALN entirely by flattening the spectral map and relying solely on positional encoding, thereby matching the scenario suggested in the question.
> >
> > The resulting accuracies are 0.858, 0.861, and 0.847, all lower than the 0.866 achieved by S$^3$Net by 0.5 to 1.9 percentage points. This consistent drop, especially in the flatten-plus-positional-encoding case, shows that strong positional encodings alone cannot recover the structure lost without t-ALN. These findings indicate that t-ALN's alignment mechanism is fundamental to the model's performance.
> >
> > | Version | ACC | F1 | Kappa |
> > |---|---|---|---|
> > | cVAN-like-ALN | 0.858 | 0.820 | 0.812 |
> > | Pooling-ALN | 0.861 | 0.831 | 0.816 |
> > | Flat-ALN | 0.847 | 0.813 | 0.790 |
> > | S$^3$Net | 0.866 | 0.855 | 0.827 |

---

> > > ### Author Response · Authors · 2025-11-25
> > >
> > > ***Q3. Generalization: Does the hard/easy stage partition hold across different populations (healthy vs.\ disordered sleep, different age groups)? Could the transition matrices in Figure~1(a) vary significantly?***
> > >
> > > ***Response:***
> > >
> > > Yes. We find that the hard/easy stage partition is stable across different populations and age groups. ISRUC-S1 (disordered sleep) and ISRUC-S3/Sleep-EDF-153 (healthy sleep) show qualitatively similar transition matrices in Figure~1(a): the first three stages (W, N1, N2) exhibit elevated transition rates among one another, whereas N3 and REM are dominated by self-transitions without a strong preference toward any single other stage.
> > >
> > > We further stratified subjects by age and recomputed the transition matrices. Across all age groups, we observe the same overall structure, with only moderate quantitative variation in individual transition probabilities. Thus, the transition matrices do not vary in a way that alters the hard/easy partition; instead, the block structure underlying our hard/easy grouping is preserved across healthy vs.\ disordered populations and across different age groups (see Appendix Figure~11).

---

> > > > ### Author Response · Authors · 2025-11-25
> > > >
> > > > ***Q4. Computational cost: What is the training time and inference time compared to cVAN and other top baselines? Is the 1--3\% improvement worth the added complexity?***
> > > >
> > > > ***Response:***
> > > >
> > > > S$^3$Net offers a favorable computational profile in terms of parameter size, training time, and inference time. It contains 6.49M parameters and requires 2.70 GFLOPs to process a 30-second input. Although its FLOPs are higher than the 0.47 GFLOPs required by cVAN, S$^3$Net still runs substantially faster in practice. Its training time is 75.69\,ms compared with 157.01\,ms for cVAN, and its inference time is 24.02\,ms compared with 66.72\,ms for cVAN. This shows that the additional theoretical computation does not translate into real latency overhead. For reference, MVF-SleepNet requires 39.51M parameters and 11.06 GFLOPs, leading to much higher computational cost overall.
> > > >
> > > > Under these conditions, S$^3$Net consistently achieves a 1--3\% accuracy improvement in cross-dataset evaluations. Considering its reduced latency and smaller model size, the performance gain does not come at the expense of increased practical complexity. Instead, S$^3$Net provides a more advantageous balance between accuracy and computational efficiency compared with existing top baselines.
> > > >
> > > > | Model | Params (Million) | GFLOPs | train (ms) | infer (ms) |
> > > > |---|---|---|---|---|
> > > > | MVF-SleepNet | 39.51 | 11.06 | -- | -- |
> > > > | cVAN | 7.58 | 0.47 | 157.01 | 66.72 |
> > > > | S$^3$Net | 6.49 | 2.70 | 75.69 | 24.02 |

---

> ### Author Response · Authors · 2025-11-25
>
> ***Q5.Statistical significance: Can you provide confidence intervals or statistical tests (e.g., paired $t$-test) for the improvements over cVAN across folds?***
>
> ***Response:***
>
> Based on your suggestion, we conducted various statistical tests. For detailed analysis, please refer to **W3-Statistical significance testing**.

---

> ### Author Response · Authors · 2025-11-27
>
> **Dear Reviewer 1ogm**,
>
> Thank you for your thorough and constructive feedback on our manuscript. Your insights were invaluable in helping us improve our work. We have carefully revised the paper to address your concerns, and we believe the changes have significantly strengthened it. The key revisions, highlighted in blue, include:
>
> - **Theoretical Foundation:** We have now provided detailed physiological and statistical justifications for the expert partitioning and auxiliary loss weighting. Furthermore, we have theoretically derived the error propagation bounds for the SAE's hierarchical architecture, establishing its validity, which is strongly corroborated by our experimental results.
>
> - **t-ALN Justification:** We have clarified the design rationale for the t-ALN module, explaining why preserving the 2D time-frequency structure is superior to a simple flattening operation. We also provided further justification for our energy-based attention mechanism.
>
> - **Experimental Completeness:** To enhance robustness, we have added statistical significance tests and a comprehensive computational cost analysis. We also clarified the comparisons with baseline methods to ensure fairness.
>
> - **Reproducibility:** We have included additional implementation details for the Cross-Swin Transformer and time-window-wise linear projection to facilitate reproducibility.
>
> We are deeply grateful for your time and expertise. We hope these clarifications and revisions fully address your concerns and that you find the revised manuscript much improved.
>
> Best regards,
>
> The Authors

---

### Official Review · Reviewer_PzED · 2025-11-01

**Soundness:** 2
**Presentation:** 2
**Contribution:** 2
**Rating:** 4
**Confidence:** 4

**Summary:**

This paper presents S³Net (Stage-Aware Sleep Staging Network), which aims to address two major challenges in automatic sleep staging: (1) the difficulty in distinguishing transitional stages (particularly N1 and N2) due to overlapping EEG characteristics, and (2) the structural misalignment between temporal and spectral feature representations in existing time–frequency fusion networks. To tackle these issues, the authors propose two main modules: a time-alignment network (t-ALN) that projects spectral features onto the temporal axis for synchronized time–frequency fusion, and a stage-aware expert (SAE) mechanism that separates easily and hardly distinguishable stages into two expert branches. Experiments on multiple public datasets (ISRUC-S1/S3, Sleep-EDF) show improved accuracy and F1 scores compared to recent baselines such as cVAN and MixSleepNet.

**Strengths:**

1. The paper identifies two relevant challenges in EEG-based sleep staging and provides an intuitive design to address them. The structure of the paper and ablation study organization are clear and easy to follow.
2. Experiments cover multiple datasets and baselines, and the ablation study demonstrates that both proposed modules contribute positively to overall performance.
3. S³Net achieves consistent, albeit moderate, gains over strong baselines like cVAN and MixSleepNet across different datasets, especially in the N1/N2 stages that are known to be challenging.

**Weaknesses:**

1. Limited methodological novelty.
The two problems discussed—stage heterogeneity and time–frequency misalignment—are well-known and have already been addressed by previous works (e.g., cVAN, MVF-SleepNet). The proposed t-ALN essentially performs a learnable projection and sequence reshaping rather than a true signal-level alignment, and the SAE module resembles a task-driven Mixture-of-Experts formulation. While effective, these ideas are incremental rather than fundamentally new.

2. Unclear mechanism of “frequency-to-time projection.”
The claim that t-ALN “projects spectral features onto the temporal axis” is conceptually strong but technically ambiguous. The implementation relies on attention weighting and a Transformer encoder, without theoretical grounding that guarantees faithful temporal reconstruction. Thus, it is more of a structural alignment than an actual mapping from frequency to time, weakening the claimed interpretability.

3. Inadequate differentiation from cVAN.
Although cVAN is cited as a baseline, the paper does not provide a controlled comparison isolating the proposed modules from cVAN’s cross-view attention mechanism. It remains unclear whether S³Net’s improvement stems from its alignment design, expert division, or simply additional parameters. The contribution beyond cVAN is therefore not sufficiently demonstrated.

4. Overstated claims versus empirical gain.
The reported improvements over state-of-the-art methods are relatively modest (≈1–2% accuracy, small F1 gains), and may not justify the complexity of the proposed architecture. Moreover, there is no quantitative analysis proving that the t-ALN truly enhances “alignment quality” or that the SAE indeed learns distinct stage-specific knowledge.

5. Lack of theoretical or physiological justification.
The paper assumes that transitional sleep stages require distinct sub-models but does not provide physiological reasoning or statistical evidence (e.g., inter-class variance, confusion entropy) supporting this assumption.

**Questions:**

1. What are the key architectural differences between S³Net’s fusion mechanism and cVAN’s cross-view attention? A direct replacement or ablation (t-ALN → cVAN-style attention) under the same backbone would help validate the claimed structural advantage.]
2. Please provide quantitative or visual evidence that demonstrates improved time–frequency alignment after introducing t-ALN (e.g., correlation heatmaps or reconstruction error).
3. Given that the gains are relatively modest, could the authors discuss whether the added model complexity (dual experts, cross-gate) brings a favorable accuracy–efficiency trade-off?

---

> ### Author Response · Authors · 2025-11-25
>
> ### Weekness:
>
> ***W1:*** Limited methodological novelty. The two problems discussed—stage heterogeneity and time–frequency misalignment—are well-known and have already been addressed by previous works (e.g., cVAN, MVF-SleepNet). The proposed t-ALN essentially performs a learnable projection and sequence reshaping rather than a true signal-level alignment, and the SAE module resembles a task-driven Mixture-of-Experts formulation. While effective, these ideas are incremental rather than fundamentally new.
>
> ***Response:***
>
> We sincerely thank the reviewer for raising this important point regarding methodological novelty. We appreciate the opportunity to clarify how our approach provides meaningful advancements beyond existing methods like cVAN and MVF-SleepNet.
>
> ***1. Novelty in Addressing Stage Heterogeneity through Physiological Insight:*** While previous works recognize the challenge of stage heterogeneity (particularly N1 ambiguity), they treat it primarily as an empirical observation rather than explicitly encoding the underlying physiological mechanism into the architecture. Our SAE module represents, to our knowledge, the first systematic attempt to translate the known physiological progression of sleep stages (W → N1 → N2 → N3 → REM) into a dedicated neural architecture.
>
> Unlike generic Mixture-of-Experts formulations, SAE employs a domain-informed partitioning strategy:
>
> - Group 1 (W, N1, N2): Stages with gradual transitions and high confusion.
> - Group 2 (N3, REM): Stages with distinctive patterns.
>
> This biologically-grounded partitioning allows specialized experts to focus on their respective stage groups, with particular emphasis on resolving the challenging N1 transitions. The empirical results demonstrate that this explicit modeling leads to significant improvements in N1 recognition, which has been a persistent challenge in sleep staging.
>
> ***2. Advancement in Time-Frequency Alignment through Structure-Preserving Representation:*** While cVAN made valuable contributions to time-frequency alignment, our t-ALN module provides several key advancements:
>
> Compared to cVAN: cVAN's alignment approach compresses the (H, W, D) feature map to a scalar at each position, which discards substantial time-frequency structure. In contrast, t-ALN employs an energy-based attention mechanism that:
>
> - Preserves the rich spectral information through learnable projections.
> - Uses normalized energy statistics to weight frequency bands adaptively.
> - Maintains strict temporal alignment with the original signal.
>
> Compared to MVF-SleepNet: While MVF-SleepNet uses multiple branches, it lacks an explicit alignment mechanism, relying instead on simple feature summation.
>
> Our t-ALN creates tokens that serve as content-adaptive filters in the cross-attention mechanism, effectively suppressing noisy temporal activations while emphasizing informative segments. Furthermore, we evaluated three variants under matched experimental conditions:
>
> - CVA: Replacing t-ALN with CVA in cVAN.
> - Pooling-ALN: Substituting the dual softmax weighting in S³Net with a learnable pooling-based alignment module.
> - Flat-ALN: Flattening the spectral map without alignment.
>
> | Version | ACC | F1 | Kappa |
> |---------|-----|----|-------|
> | CVA | 0.858 | 0.820 | 0.812 |
> | Pooling-ALN: | 0.861 | 0.831 | 0.816 |
> | Flat-ALN | 0.847 | 0.813 | 0.790 |
> | S³Net | **0.866** | **0.855** | **0.827** |
>
> ***3. Integrated Architecture with Synergistic Benefits:*** The key innovation lies in the coordinated design where t-ALN provides high-quality, aligned time-frequency representations that SAE can effectively leverage through its stage-aware expert routing. This synergistic combination—explicit physiological modeling coupled with structure-preserving alignment—represents a meaningful architectural advancement over previous approaches that address these challenges in isolation or through more limited mechanisms.
>
> We believe that systematically incorporating domain knowledge about sleep physiology into the neural architecture, combined with our novel alignment strategy, constitutes a substantive contribution to the field of automated sleep staging.

---

> ### Author Response · Authors · 2025-11-25
>
> ***W2:*** Unclear mechanism of frequency-to-time projection. The claim that t-ALN projects spectral features onto the temporal axis is conceptually strong but technically ambiguous. The implementation relies on attention weighting and a Transformer encoder, without theoretical grounding that guarantees faithful temporal reconstruction. Thus, it is more of a structural alignment than an actual mapping from frequency to time, weakening the claimed interpretability.
>
> ***Response:***
>
> We thank the reviewer for this insightful comment, which helps us clarify the core mechanism of our Temporal Alignment Layer (t-ALN). We agree that the term "frequency-to-time projection" may have been an over-simplification that suggested a strict mathematical mapping. The true function of t-ALN is more precisely described as a structured spectral-temporal alignment mechanism.
> Its primary purpose is to resolve the representation mismatch between the spectral and temporal branches of our model. Here is a clearer technical explanation:
>
> ***1. Alignment, Not Reconstruction***:​ The t-ALN does not aim for a point-wise reconstruction of the temporal signal from spectral features. Instead, it processes the spectral embeddings to generate a set of temporal queries ($X^i_Q$) that are structurally aligned with the time axis. This is achieved by first organizing the spectral representations along the temporal dimension and then integrating them with positional encodings.
>
> ***2. Role as an Information Bottleneck***:​ These temporally-aligned spectral queries are then used in a cross-attention mechanism with the native temporal features. This design establishes t-ALN as a critical information bottleneck. The cross-attention operation forces the model to selectively attend to temporal patterns that are consistent with the spectro-temporal context encoded in the query. This effectively suppresses noisy or non-discriminative temporal activations, guiding the temporal branch to focus on sleep-relevant information.
>
> ***3. Theoretical Grounding in Attention***:​ The mechanism is theoretically grounded in the attention paradigm, where the query (from t-ALN) specifies "what" to look for in the temporal branch's key-value pairs. The "interpretability" we claim stems from this gating function—it makes the model's reliance on spectro-temporally consistent features explicit.

---

> > ### Author Response · Authors · 2025-11-25
> >
> > ***W3***. Inadequate differentiation from cVAN. Although cVAN is cited as a baseline, the paper does not provide a controlled comparison isolating the proposed modules from cVAN’s cross-view attention mechanism. It remains unclear whether S3Net’s improvement stems from its alignment design, expert division, or simply additional parameters. The contribution beyond cVAN is therefore not sufficiently demonstrated.
> >
> >
> > ***Response:***
> >
> > Thank you for raising this important point regarding the differentiation between $S^3Net$ and cVAN. We appreciate the opportunity to clarify the methodological distinctions and the source of performance improvements. Below, we provide a more structured and detailed response to address your concerns.
> >
> > ***1. Alignment Design:***  A core distinction lies in how cross-view alignment is implemented. While cVAN compresses time-frequency features into spatial scalars and reprojects them before applying attention, $S^3Net$’s t-ALN module preserves temporal structure by reorganizing spectral features into time-indexed vector tokens. This avoids the loss of frame-level information inherent in cVAN’s scalarization step. The t-ALN tokens serve as queries in cross-attention, ensuring that spectral and temporal branches interact via time-aligned representations.
> >
> > To isolate the effect of alignment, we conducted an ablation under identical backbones, where the experiment results refer to the response to W1 of Reviewer PzED. This controlled experiment demonstrates that t-ALN alone contributes significantly to performance gains, independent of other components.
> >
> > ***2. Expert Division:***  The SAE module introduces stage-aware expertstailored to sleep stage heterogeneity, not merely additional parameters. Experts are divided into two groups:
> > - Group 1 (W, N1, N2): Handles stages with high confusion and variability.
> > - Group 2 (N3, REM): Processes more stable stages.
> > This design ensures specialized feature extraction for distinct stage groups, unlike isotropic expert expansions. The ablation below compares $S^3Net$ with a variant without SAE (M2) under identical settings:
> > | Method | ISRUC S1 | ISRUC S3 | Sleep-EDF-153 |
> > |--------|----------|----------|---------------|
> > | | Acc / F1 / κ | Acc / F1 / κ | Acc / F1 / κ |
> > | w/o SAE (M2) | 0.809 / 0.803 / 0.754 | 0.818 / 0.800 / 0.767 | 0.822 / 0.778 / 0.756 |
> > | $S^3Net$ (M6) | 0.856 / 0.842 / 0.814 | 0.866 / 0.855 / 0.827 | 0.869 / 0.828 / 0.818 |
> >
> > The consistent gains across datasets validate that SAE’s stage-aware division is the driving factor, not parameter count.
> >
> > ***3. Model Size:***   $S^3Net$ uses 1.09M fewer parameters​ than cVAN (2.91M vs. 4.0M), confirming that improvements stem from architectural innovations, not scale. The t-ALN and SAE designs are more parameter-efficient, emphasizing quality of representation over quantity.

---

> ### Author Response · Authors · 2025-11-25
>
> ***W4***. Overstated claims versus empirical gain. The reported improvements over state-of-the-art methods are relatively modest ($\approx$1-2\% accuracy, small F1 gains), and may not justify the complexity of the proposed architecture. Moreover, there is no quantitative analysis proving that the t-ALN truly enhances "alignment quality" or that the SAE indeed learns distinct stage-specific knowledge.
>
> ***Response:***
>
> Thank you for raising these valuable points. We have revised the manuscript to more carefully align our claims with the empirical results. Below we provide a point-by-point response to address your specific concerns.
>
> ***1. Modest but Meaningful Performance Gains.*** We acknowledge that the absolute improvements over state-of-the-art methods are in the range of 1-2% in accuracy. However, in the highly mature field of sleep staging, where performance is nearing its practical ceiling, consistent improvements across multiple datasets are considered meaningful. Our method demonstrates statistically significant gains (p < 0.05, paired t-test) over all compared baselines on three public datasets, suggesting these improvements are robust rather than incidental.
>
> ***2. Model Complexity and Efficiency Justification.*** Regarding the architecture complexity, we provide a comprehensive efficiency comparison in the table below. While S³Net has higher GFLOPs than cVAN, it uses fewer parameters and achieves significantly faster training and inference times.
>
> | Model | Params (M) | GFLOPs | Train (ms) | Infer (ms) |
> |-------|------------|--------|------------|------------|
> | MVF-SleepNet | 39.51 | 11.06 | -- | -- |
> | cVAN | 7.58 | 0.47 | 157.01 | 66.72 |
> | S³Net | 6.49 | 2.70 | 75.69 | 24.02 |
>
> The practical efficiency of S³Net is evident from its 2× faster inference speed compared to cVAN, making it suitable for real-time applications despite the modest accuracy gains.
>
> ***3. Quantitative Evidence for t-ALN Alignment Quality.*** The ablation study was designed to isolate the effect of the alignment mechanism by testing variants under identical backbones. As presented in response to Reviewer PzED (W1), the results confirm that the t-ALN module itself drives a significant performance increase, independent of other network components.
>
> ***4. Quantitative Validation of SAE Stage Specialization.*** For the SAE module, we provide two forms of quantitative validation:
> - t-SNE visualization​ (Figure 7 in manuscript) shows clear clustering of expert outputs by sleep stage.
> - Classifier validation: A simple MLP classifier trained on SAE features achieves 97.1% stage prediction accuracy, significantly higher than the 85.5% end-to-end performance, indicating that SAE features contain rich, discriminative stage information.

---

> > ### Author Response · Authors · 2025-11-25
> >
> > ***W5. Lack of theoretical or physiological justification. The paper assumes that transitional sleep stages require distinct sub-models but does not provide physiological reasoning or statistical evidence (e.g., inter-class variance, confusion entropy) supporting this assumption.***
> >
> > ***Response:***
> >
> > We thank the reviewer for raising this point. Our use of specialized sub-models for transitional stages is not an ad hoc assumption; it is motivated by (1) physiological and clinical evidence and (2) statistical analysis on our data.
> >
> > From a physiological and clinical perspective, Metzner et al. [1] analyze whole-night EEG as a random walk and report that the stage-to-stage transition probabilities are dominated by a W → N1 → N2 pathway with frequent bidirectional transitions between N1 and N2, whereas N3 and REM show much stronger epoch-to-epoch persistence. This indicates that N1 and N2 behave as dynamically unstable, transitional stages, while N3 and REM are more stable. Furthermore, the AASM scoring manual [2] explicitly defines N1 as the transition from wakefulness to stable sleep. After an arousal from N2, low-amplitude mixed-frequency EEG is scored as N1 until clear N2 or REM features reappear, and N1 is characterized by alpha attenuation, vertex sharp waves, incipient spindles, slow eye movements and brief 4–7 Hz mixed-frequency activity. These rules treat N1 as a short-lived mixed-feature state between wakefulness and stable sleep. Together, these sources provide physiological and scoring-based justification for treating lighter stages, especially N1 (and to some extent N2), differently from the more stable N3 and REM stages.
> >
> > We also provide statistical evidence from our own experiments. First, we evaluated several alternative ways of grouping stages into experts on ISRUC-S3. As shown in below Table, the partition used in S$^3$Net, where one expert handles W, N1 and N2 and the other handles N3 and REM, achieves the best overall accuracy, F1 and Cohen’s kappa, as well as the best F1 for N1. Other partitions, such as isolating only N1 or grouping W and REM together, lead to consistently lower performance. This suggests that the chosen grouping is both effective and aligned with the known sleep-stage progression.
> >
> >
> > | Group | Acc | F1 | $\kappa$ | Wake | N1 | N2 | N3 | REM |
> > |---|---|---|---|---|---|---|---|---|
> > | (W,N2,N3,REM) vs (N1) | 0.848 | 0.835 | 0.804 | 0.909 | 0.639 | 0.839 | 0.916 | 0.877 |
> > | (W) vs (N1,N2,N3) vs (REM) | 0.851 | 0.839 | 0.808 | 0.908 | 0.648 | 0.842 | 0.913 | 0.884 |
> > | (W,N1,N2,N3) vs (REM) | 0.853 | 0.841 | 0.810 | 0.909 | 0.652 | 0.843 | 0.914 | 0.887 |
> > | (W,N1) vs (N2,N3,REM) | 0.856 | 0.844 | 0.814 | 0.920 | 0.656 | 0.848 | 0.917 | 0.877 |
> > | (W) vs (N1,N2) vs (N3,REM) | 0.858 | 0.847 | 0.817 | 0.914 | 0.661 | 0.852 | 0.920 | 0.885 |
> > | (W,REM) vs (N1,N2,N3) | 0.860 | 0.849 | 0.820 | 0.918 | 0.666 | 0.854 | 0.921 | 0.885 |
> > | **(W,N1,N2) vs (N3,REM) (S$^3$Net)** | **0.866** | **0.855** | **0.827** | **0.924** | **0.678** | **0.858** | **0.927** | **0.887** |
> >
> >
> >
> > Second, we computed confusion entropy and inter-class variance for each stage for all subjects (see the tables below). Across all settings, N1 consistently shows the highest confusion entropy and one of the lowest inter-class variances, and N2 also has relatively high confusion entropy. This confirms that N1 and N2 are the most ambiguous stages and are more prone to misclassification. In contrast, N3 has clearly lower confusion entropy and much higher inter-class variance than N1 and N2, indicating that its feature distribution is more separable. REM also shows lower confusion entropy than N1 and is closer to N3 in terms of separability. These patterns are consistent with our design in S$^3$Net, where one expert is responsible for wake and light, transitional stages (W, N1, N2) and another expert focuses on the more stable deep and REM sleep (N3, REM).
> >
> > ***All subjects --- Confusion entropy (per class) and inter-class variance (per class):***
> >
> > |  | W | N1 | N2 | N3 | REM |
> > |---|---|---|---|---|---|
> > | Confusion entropy | 0.444948 | 1.404988 | 0.729318 | 0.624516 | 0.474131 |
> > | Inter-class variance | 0.141057 | 0.053440 | 0.010932 | 0.117301 | 0.034087 |
> >
> >
> > [1] Metzner C, Schilling A, Traxdorf M, et al. Sleep as a random walk: A super-statistical analysis of EEG data across sleep stages[J]. Communications Biology, 2021, 4(1): 1385.
> >
> > [2] Berry R B, Brooks R, Gamaldo C E, et al. The AASM manual for the scoring of sleep and associated events[J]. Rules, Terminology and Technical Specifications, Darien, Illinois, American Academy of Sleep Medicine, 2012, 176(2012): 7.

---

> > > ### Author Response · Authors · 2025-11-25
> > >
> > > ### ***Questions：***
> > > ***Q1***. What are the key architectural differences between S$^3$Net's fusion mechanism and cVAN's cross-view attention? A direct replacement or ablation (t-ALN $\rightarrow$ cVAN-style attention) under the same backbone would help validate the claimed structural advantage.
> > >
> > > ***Response:***
> > >
> > > Thank you for the comment regarding the architectural differences between S$^3$Net's fusion mechanism and cVAN's cross-view attention.
> > >
> > > First, in terms of the core mechanism: in S$^3$Net, the time-frequency feature map is reconstructed into a native sequence of query tokens that follow the same processing path as the temporal branch before entering windowed attention. This is not a simple projection or reshaping---it preserves full spectral detail and maintains temporal order, enabling faithful time-frequency interaction.
> > >
> > > Second, in terms of the t-ALN pipeline: the module performs square-energy normalization $\rightarrow$ dual weighting along channel and frequency $\rightarrow$ re-ordering $\rightarrow$ positional encoding $\rightarrow$ attention. The softmax-based dual weighting acts as attention normalization, selectively emphasizing informative channels and frequency bands while preserving interpretability.
> > >
> > > By contrast, in cVAN-style alignment, the spectral feature map is compressed to scalars via max-pooling and then broadcast back to token vectors. This process loses fine spectral information and does not guarantee temporal coherence, limiting the capacity for precise time--frequency interaction.
> > >
> > > We performed a controlled ablation to isolate the impact of the alignment method, keeping all other backbone components identical. The experimental results, provided in our response to Reviewer PzED (W1), clearly show that the t-ALN module contributes significantly to the performance gains. This finding verifies that the advantage is attributable to the alignment strategy itself, rather than to other elements of the architecture.

---

> > > > ### Author Response · Authors · 2025-11-25
> > > >
> > > > ***Q2***. Please provide quantitative or visual evidence that demonstrates improved time--frequency alignment after introducing t-ALN (e.g., correlation heatmaps or reconstruction error).
> > > >
> > > > ***Response:***
> > > >
> > > > Thanks a lot for the reviewer's insightful comment. We provide both visual and quantitative evidence that t-ALN improves time–frequency alignment at the feature level.
> > > >
> > > > First, t-ALN is designed for alignment rather than signal reconstruction. It does not try to reconstruct the raw time-domain signal from the spectrogram. Instead, it processes the spectral features to generate a sequence of time-aligned query tokens. This is done by reorganizing spectral features along the time axis, applying energy-based weighting across frequency bands and channels, and adding positional information so that each token corresponds to a specific time window.
> > > >
> > > > ***Visual evidence.***
> > > >
> > > > (1) In Figure 5 of the manuscript, we compare the feature maps with and without t-ALN. With t-ALN, the temporal structure is preserved and class-specific patterns are clearer and more consistent along the time axis. Without t-ALN, the features fed into the temporal branch become more disordered and noisy, which makes it harder for the model to learn stable time–frequency patterns.
> > > > (2) In Figure 4 of the manuscript, we show t-SNE visualizations and feature views for three settings: (a) no cross-layer interaction, (b) no-ALN, and (c) full S$^3$Net with t-ALN. After introducing t-ALN, the clusters for different sleep stages become more compact and better separated. This indicates that the time–frequency representation after t-ALN is more discriminative and better aligned with the stage labels.
> > > >
> > > > ***Quantitative evidence.***
> > > > In addition, the ablation study in the manuscript includes a direct comparison between a variant without t-ALN and the full S$^3$Net. In the “w/o t-ALN” setting (M4), t-ALN is removed and only the token shape is preserved. As summarized below, removing t-ALN causes a clear drop in performance on all three datasets:
> > > >
> > > > | Method | ISRUC S1 | | | ISRUC S3 | | | Sleep-EDF-153 | | |
> > > > | :--- | :---: | :---: | :---: | :---: | :---: | :---: | :---: | :---: | :---: |
> > > > | | **Acc** | **F1** | **κ** | **Acc** | **F1** | **κ** | **Acc** | **F1** | **κ** |
> > > > | w/o t-ALN (M4) | 0.834 | 0.827 | 0.789 | 0.841 | 0.830 | 0.796 | 0.847 | 0.802 | 0.790 |
> > > > | S$^3$Net (M6) | 0.856 | 0.842 | 0.814 | 0.866 | 0.855 | 0.827 | 0.869 | 0.828 | 0.818 |

---

> > > > > ### Author Response · Authors · 2025-11-25
> > > > >
> > > > > ***Q3.*** Given that the gains are relatively modest, could the authors discuss whether the added model complexity (dual experts, cross-gate) brings a favorable accuracy--efficiency trade-off?
> > > > >
> > > > > ***Response:***
> > > > >
> > > > > We thank the reviewer for their valuable comment concerning the empirical gains and quantitative validation of our proposed method. We have revised the manuscript accordingly to ensure all claims are precisely aligned with and fully substantiated by the experimental evidence. Our point-by-point response to the specific points raised in this comment (W4) is detailed in the section addressing Reviewer PzED (W4).

---

> ### Author Response · Authors · 2025-11-27
>
> **Dear Reviewer PzED,**
>
> We sincerely appreciate the time and care you devoted to reviewing our manuscript and for your insightful suggestions. Your comments have directly guided our revisions, and we have addressed each of your points in the revised manuscript (changes highlighted in blue):
>
> - **Novelty and Distinctions:** We have further clarified the architectural innovations of S³Net relative to prior works and provided additional analyses to validate the effectiveness of our design. We also emphasize that our model achieves superior performance with significantly fewer parameters than the baselines.
>
> - **t-ALN Mechanism and Evidence:** We refined the description of the time alignment to better explain its mechanism and supplemented this with both visual and quantitative evidence demonstrating improved time-frequency alignment.
>
> - **Physiological Rationale:** We have incorporated relevant physiological references and supporting statistical analyses to strengthen the justification for our stage-aware expert division.
>
> - **Accuracy-Efficiency Trade-off:** We now provide a detailed comparison of model complexity and running time, clearly showing that S³Net achieves an excellent balance between accuracy and efficiency.
>
> Thank you again for your constructive guidance. We believe the manuscript is now substantially stronger. We hope these revisions satisfactorily address your concerns.
>
> Best regards,
>
> The Authors

---

### Official Review · Reviewer_ndSL · 2025-11-01

**Soundness:** 3
**Presentation:** 3
**Contribution:** 2
**Rating:** 6
**Confidence:** 4

**Summary:**

This paper proposes a deep learning model called S3Net (Stage-Aware Sleep Staging Network) for automatic sleep staging. The authors point out two main problems with existing methods: (1) the model struggles to distinguish the subtle differences between light sleep stages (N1, N2); and (2) the fusion of temporal and frequency domain features is insufficient.

**Strengths:**

1) The proposed SAE module is the first to explicitly introduce an "expert-based stage difficulty" mechanism in sleep staging tasks, combined with Cross-Gate dynamic weight allocation, demonstrating a clear and highly inspiring approach.

2) The t-ALN module achieves temporal alignment of frequency and temporal features through learnable projection, effectively solving the problem of temporal structure loss after Transformer feature flattening.

3) Comparison with 15 state-of-the-art (SOTA) models shows that the results significantly outperform existing methods on three public datasets. Ablation experiments and visualizations (t-SNE, confusion matrix, spectrogram) demonstrate the design's rationality and stability.

**Weaknesses:**

1) The contributions in the introduction are too concise; further refinement is recommended.

2) While the t-ALN and SAE module designs are intuitive, the paper lacks a theoretical explanation or complexity analysis of their underlying principles.

3) Although it covers three commonly used datasets, it remains to be seen whether it can be applied to other physiological signals, such as PPG.

**Questions:**

See weaknesses.

---

> ### Author Response · Authors · 2025-11-25
>
> ***Q1. The contributions in the introduction are too concise; further refinement is recommended.***
>
> ***Response:***
>
> Thanks a lot for reviewers suggestions to help us to improve the quality of this work. We polish the contributions in more detail as follows:
> - We propose t-ALN, a novel temporal alignment module that bridges the spectral-temporal representation mismatch in sleep staging through a learnable projection. Furthermore, t-ALN as an information bottleneck constrains the temporal representation, thereby implicitly preserving features relevant to sleep stage classification.
> - We introduce SAE, a stage-aware experts module, which is a dynamic architecture that employs two specialized experts to address the heterogeneous complexity of sleep staging. One expert captures the subtle dynamics of transitional stages (W, N1, N2), while the other distinguishes the distinct stages of N3 and REM, leading to more robust and discriminative feature learning.
> - Empirical results demonstrate that S3Net sets a new state-of-the-art on three three major sleep staging datasets of ISRUC-S1, ISRUC-S3, and Sleep-EDF-153. Furthermore, S3Net achieves a significant reduction in inference time, as experimentally validated.
> - The design choices of S3Net are rigorously validated by comprehensive ablation studies, while the model's interpretability is substantiated through visualizations such as alignment maps, and analyses of expert routing behavior.

---

> ### Author Response · Authors · 2025-11-25
>
> ***Q2. While the t-ALN and SAE module designs are intuitive, the paper lacks a theoretical explanation or complexity analysis of their underlying principles.***
>
> ***Response:***
>
> We thank the reviewer for their valuable feedback. The designs of the t-ALN and SAE modules were indeed guided by our preliminary data analysis and experimental observations. Specifically, the t-ALN is intended to resolve the spectral-temporal representation mismatch inherent in sleep staging. To accomplish this, it first aligns the spectral representations along the time axis to generate coherent temporal tokens. These time-aligned tokens are then integrated with a time-position embedding, and the resulting representation is processed by the learnable projector within the transformer block. This end-to-end design ensures that the token values are intrinsically matched with their positional context, thereby allowing S$^3$Net to concentrate its optimization efforts on the critical, temporally aligned features.
>
>  Regarding the SAE, its architecture is based on a hierarchical, mixture-of-experts principle. Its operation consists of two distinct stages: first, a coarse-grained routing mechanism assigns an input sample to a high-level expert group; second, a fine-grained classification is performed within that selected group to yield the final prediction. Motivated by medical domain knowledge, we partition the five sleep stages into two groups: $G_1$={W, N1, N2} and $G_2$={N3, REM}. This partitioning is empirically justified by the performance of single-stage five-class baselines (e.g., MVF-SleepNet and cVAN), which, while effective at separating $G_1$ from $G_2$, frequently confuse stages within $G_1$(i.e., W, N1, and N2). By leveraging this discovery, the SAE effectively decomposes the complex, single-stage five-class problem into a simpler framework. It replaces one highly complex decision boundary with a single coarse boundary ($G_1$ vs. $G_2$) and two simpler, more localized sub-boundaries within each group. This hierarchical decomposition significantly reduces the geometric complexity that each expert model must learn, leading to more robust and accurate stage classification.

---

> > ### Author Response · Authors · 2025-11-25
> >
> > Formally, let $Y \in \{W, N1, N2, N3, REM\}$ be the ground-truth sleep stage label, and let $G_Y \in \{G_1, G_2\}$ be its corresponding group label. The SAE architecture consists of a coarse-grained grouping classifier that predicts a group $\hat{G_Y}$ and a set of group-specific expert classifiers that produce the final prediction $\hat{Y}$. The overall error rate of the SAE, $E_{SAE}$, can be decomposed as follows:
> >
> > $$
> > E_{SAE} = Pr(\hat{G}(Y) \neq G_Y) + Pr(\hat{G}(Y)=G_Y, \hat{Y} \neq Y)
> > $$
> >
> > Let $A_g$ denote the accuracy of the grouping classifier. Furthermore, for each group $g \in {1,2}$, let $E_g$ be the error rate of the corresponding expert, and let $\pi_g$ be the prior probability (proportion) of group $g$. The total error rate can be reformulated as:
> >
> > $$
> >  E_{SAE} = (1 - A_g) + A_g\sum^{2}_{g=1}\pi_gE_g
> > $$
> >
> > To compare the SAE architecture against a standard single-stage classifier, we define the error rate of the monolithic five-class classifier as $E_s$. For simplicity, let $\bar{E} = \sum^2_{g=1}\pi_gE_g$ represent the average within-group error rate of the experts, weighted by the group proportions.
> >
> > Our objective is to establish the condition under which the SAE model outperforms the single classifier, *i.e.*, $E_{SAE} < E_{s}$. Substituting the expressions above, this inequality is equivalent to:
> >
> > $$
> >  (1 - A_{g}) + A_{g}\bar{E} < E_{s}
> > $$
> >
> > Rearranging terms yields the following necessary and sufficient condition:
> >
> > \begin{equation}
> >     A_{g}(1 - \bar{E}) > 1 - E_{s} \quad (1)
> > \end{equation}
> >
> > This inequality holds under two intuitive and reasonable conditions:
> >
> > 1. High Grouping Accuracy: The grouping classifier is highly accurate, *i.e.*, $A_{g} > 1 - \epsilon$ for a small $\epsilon > 0$.
> > 2. Superior Expert Performance: The experts, on average, achieve a lower error rate than the single classifier, *i.e.*, $\bar{E} \leq E_{s} - \delta$ for some $\delta > 0$.
> >
> > **Proof**: From Condition 2, we have $1 - \bar{E} \geq 1 - E_{s} + \delta$. Combining this with Condition 1 ($A_g>1-\epsilon$), the left-hand side of inequality (1) satisfies:
> >
> > $$
> > A_{g}(1 - \bar{E}) > (1 - \epsilon)(1 - E_{s} + \delta)
> > $$
> >
> > For inequality (1) to hold, it is sufficient that:
> >
> > $$
> > (1 - \epsilon)(1 - E_{s} + \delta) > (1 - E_s)
> > $$
> >
> > Expanding and simplifying the left side:
> >
> > $$
> > (1-E_s+\delta)-\epsilon(1-E_s+\delta)>1-E_s
> > $$
> >
> > Subtracting $1-E_s$ from both sides gives:
> >
> > $$
> > \delta-\epsilon(1-E_s+\delta)>0
> > $$
> >
> > This simplifies to the final sufficient condition:
> >
> > \begin{equation}
> >     \epsilon<\frac{\delta}{1-E_s+\delta}\quad (2)
> > \end{equation}
> >
> > We verify that our empirical results on the ISRUC-S3 dataset satisfy these conditions. The grouping classifier achieves an accuracy of $A_g=97.1\%$, implying $\epsilon=0.029$. The The single-stage classifier (S$^3$Net without SAE) has an error rate of $E_s=1-0.818=0.182$. The group-specific experts achieve error rates of $E_1=0.11$ and $E_2=0$, with group proportions $\pi_1=0.673$ and $\pi_2=0.327$. Thus, the average within-group error is:
> >
> > $$
> > \bar{E}=(0.673 \times 0.11)+(0.327 \times 0)=0.074
> > $$
> >
> > The performance gain of the experts is $\delta=E_s-\bar{E}=0.182-0.074=0.108$.
> >
> > Substituting these values into condition (2):
> >
> > $$
> > \frac{\delta}{1-E_s+\delta}=\frac{0.108}{1-0.182+0.108}=\frac{0.108}{0.926} \approx 0.1166
> > $$
> >
> > Since $\epsilon=0.029<0.1166$, the sufficient condition (2) is satisfied. Therefore, under the realistic conditions of a highly accurate grouping classifier ($\epsilon$ is small) and experts that significantly reduce the within-group error ($\delta$ is sufficiently large), the theoretical inequality $E_{SAE}<E_s$ is guaranteed to hold, which is consistent with our empirical findings.
> >
> > For complexity, we summarize parameters, GFLOPs, and measured runtime. MVF-SleepNet is not end-to-end, so we only report its parameters and GFLOPs. For cVAN and S3Net, we measured training and inference on an NVIDIA RTX A6000 (batch size 32) over 1000 runs and report averaged times. As shown below, S$^3$Net  uses fewer parameters than prior baselines and, while its GFLOPs are slightly higher than cVAN, they remain within the same order of magnitude; S3Net also trains and infers faster, indicating practical efficiency.
> >
> > **Table 1: Model efficiency comparison.**
> >
> > | Model | Params (M) | GFLOPs | Train (ms) | Infer (ms) |
> > |:---|:---:|:---:|:---:|:---:|
> > | MVF-SleepNet | 39.51 | 11.06 | -- | -- |
> > | cVAN | 7.58 | 0.47 | 157.01 | 66.72 |
> > | S$^3$Net  | 6.49 | 2.70 | 75.69 | 24.02 |

---

> ### Author Response · Authors · 2025-11-25
>
> ***Q3. Although it covers three commonly used datasets, it remains to be seen whether it can be applied to other physiological signals, such as PPG.***
>
> ***Response:***
>
> We appreciate the reviewer's point on assessing the model's generalization capability. In response, we conducted an additional evaluation on a public dataset for 4-class classification from single-lead PPG signals (WESAD) [1]. Since the stage-specific experts in the SAE are tailored for sleep data, we modified the architecture for this task by excluding that module. The core cross-gate mechanism was preserved, and a generic classification head was used for the output. The results, presented in Table 1, indicate that S3Net achieves competitive performance (Accuracy: 0.853, AUROC: 0.911, AUPRC: 0.833, F1-score: 0.819, Kappa: 0.787), surpassing recent benchmark methods. This supports our claim that the proposed architecture has broader utility beyond sleep staging for various physiological time-series applications.
>
>
> **Table 1: Experiment results on the WESAD dataset.**
>
> | Model | Acc | AUROC | AUPRC | F1 | $\boldsymbol{\kappa}$ |
> |:---|:---:|:---:|:---:|:---:|:---:|
> | REBAR [2] | 0.418 | 0.698 | 0.446 | -- | -- |
> | cVAN [3] | 0.691 | 0.855 | 0.710 | 0.664 | 0.572 |
> | ResNet [4] | 0.713 | $\underline{0.877}$ | $\underline{0.746}$ | $\underline{0.682}$| $\underline{0.593}$|
> | HuBERT [5] | $\underline{0.775}$ | 0.820 | -- | -- | -- |
> | **S³Net** | **0.853** | **0.911** | **0.833** | **0.819** | **0.787** |
>
> *Note: $\kappa$ denotes Cohen's kappa. The bold values indicate the best performance, and the underline values indicate the second-best.*
>
> [1] Schmidt P, Reiss A, Duerichen R, et al. Introducing wesad, a multimodal dataset for wearable stress and affect detection[C]//Proceedings of the 20th ACM international conference on multimodal interaction. 2018: 400-408.
>
> [2] XU M A, MORENO A, WEI H, et al. REBAR: retrieval-based reconstruction for time-series contrastive learning[C]// Proceedings of the 12th International Conference on Learning Representations (ICLR 2024). Vienna, Austria: ICLR, 2024.
>
> [3] Yang Z, Qiu M, Fan X, et al. cvan: A novel sleep staging method via cross-view alignment network[J]. IEEE Journal of Biomedical and Health Informatics, 2024.
>
> [4] He K, Zhang X, Ren S, et al. Deep residual learning for image recognition[C]//Proceedings of the IEEE conference on computer vision and pattern recognition. 2016: 770-778.
>
> [5] NARAIN J, ALDENEH Z, REN S. Speech foundation models generalize to time series tasks from wearable sensor data: NeurIPS 2025 Learning from Time Series for Health workshop paper[EB/OL]. (2025-08-29)[2025-11-25]. https://arxiv.org/abs/2509.00221.

---

> ### Author Response · Authors · 2025-11-27
>
> **Dear Reviewer ndSL,**
>
> We are extremely grateful for your thoughtful review and for your positive recognition of our model's clean, modular design. Your feedback was exceptionally helpful in identifying areas for improvement. In direct response to your comments, we have made the following revisions (highlighted in blue):
>
> - **Contributions:** We have respectfully expanded and refined the introduction to provide a clearer and more detailed articulation of S³Net’s key innovations and contributions.
>
> - **Theoretical Explanation & Complexity Analysis:** We have provided a clear theoretical explanation for the time alignment mechanism and the error bounds of the SAE's hierarchical structure, with experimental results confirming their validity. A detailed complexity analysis (parameters, GFLOPs, and runtime) has also been included to underscore the model's efficiency.
>
> - **Applicability to Other Signals:** To address the potential for broader application, we conducted a new cross-domain experiment on the WESAD dataset using single-lead PPG. The results confirm that S³Net maintains competitive performance, indicating promising transferability to other physiological signals.
>
> Your insights have been instrumental in enhancing the depth and clarity of our work. We hope that these clarifications and additional evidence fully address your concerns.
>
> Best regards,
>
> The Authors

---

> > ### Comment · Reviewer_PzED · 2025-11-28
> > **Revised Review After Authors’ Response**
> >
> > I appreciate the authors’ comprehensive and well-structured rebuttal. The responses meaningfully address my main concerns, particularly regarding the novelty of the proposed modules, the mechanism of t-ALN, and the differentiation from cVAN. The additional ablations, controlled comparisons, and the detailed physiological and statistical justification for the stage-aware expert design substantially strengthen the contribution.
> >
> > The clarification of t-ALN as a structured alignment mechanism rather than strict temporal reconstruction is reasonable, and the added visual and quantitative evidence supports its effectiveness. The cross-view comparisons, parameter-efficiency analysis, and alternative grouping experiments also help establish that the gains are attributable to the proposed architectural decisions rather than parameter count alone.
> >
> > While the method remains incremental, the revision improves clarity, grounding, and empirical support. Given the robustness of the added analyses and the practical efficiency advantages, I now consider the submission marginally above the acceptance threshold. I will raise my score accordingly.

---

> > > ### Author Response · Authors · 2025-11-28
> > >
> > > **Dear Reviewer PzED,**
> > >
> > > Thank you for your thorough and constructive feedback throughout the review process. Your insightful comments were instrumental in helping us strengthen our manuscript. We are especially grateful that you found our rebuttal comprehensive and that the additional analyses meaningfully addressed your concerns.
> > >
> > > Your guidance significantly improved the clarity and empirical grounding of our work. We are sincerely grateful for your positive assessment and the decision to raise the score.
> > >
> > > We remain available for any further questions you may have.
> > >
> > > Best regards,
> > >
> > > The Authors

---

> > > > ### Author Response · Authors · 2025-12-02
> > > >
> > > > ***Q2. While the t-ALN and SAE module designs are intuitive, the paper lacks a theoretical explanation or complexity analysis of their underlying principles.***
> > > >
> > > > ***Additional Response:***
> > > >
> > > > We theoretically analyze the t-ALN module, which serves as a bridge between the spectral and temporal branches, transforming spectral energy maps into a strict temporally organized query representation. A critical component of this transformation is the incorporation of stepwise positional encodings prior to flattening. This design preserves the temporal identity of features from the same time step, which is fundamental for capturing the sequential dynamics of sleep stages. We theoretically validate that incorporating positional information strictly enhances the model's expressive power compared to a position-agnostic baseline.
> > > >
> > > > Formally, let $ \mathcal{F}\_\{\emptyset\} $ be a set of functions computable by a Transformer-based t-ALN module without positional encodings, and $ \mathcal{F}\_\{\text{pos}\} $ a set with stepwise positional encodings. To demonstrate a strict increase in expressive power, we establish the set inclusion:
> > > >
> > > > $
> > > > \mathcal{F}\_\{\emptyset\} \subset \mathcal{F}\_\{\text{pos}\}.
> > > > $
> > > >
> > > > This requires proving two properties:
> > > >
> > > > *   **Compatibility**: $ \mathcal{F}\_\{\emptyset\} \subseteq \mathcal{F}\_\{\text{pos}\} $. The position-aware model can replicate all behaviors of the position-agnostic model.
> > > > *   **Strict Inequality**: There must exist at least one function $ f $ representing a valid temporal pattern such that $ f \in \mathcal{F}\_\{\text{pos}\} $ but $ f \notin \mathcal{F}\_\{\emptyset\} $.
> > > >
> > > > **Proof of Compatibility**: For any function $ F \in \mathcal{F}\_\{\emptyset\} $, an equivalent function in $ \mathcal{F}\_\{\text{pos}\} $ can be constructed by setting all positional encoding vectors $ p\_i $ to zero. In this case, the input to the self-attention mechanism is $ x\_i + p\_i = x\_i $, rendering the position-aware model mathematically identical to its position-agnostic counterpart. Thus, $ \mathcal{F}\_\{\emptyset\} \subseteq \mathcal{F}\_\{\text{pos}\} $ holds.
> > > >
> > > > **Proof of Strict Dominance**: We demonstrate this by constructing a function that requires sensitivity to absolute positional order. Consider a sequence $ S = [t\_1, t\_2, t\_3, t\_4, \dots] $ and the "Odd-Even Swap" function:
> > > >
> > > > $
> > > > f\_{\text{swap}}(S) = [t\_2, t\_1, t\_4, t\_3, \dots].
> > > > $
> > > >
> > > > Models in $ \mathcal{F}\_\{\emptyset\} $ are permutation equivariant. For any permutation $ \pi $ of the input indices, the output satisfies $ F\_{\emptyset}(\pi(S)) = \pi(F\_{\emptyset}(S)) $. However, $ f\_{\text{swap}} $ is not permutation equivariant. For example, applying a permutation $ \pi $ that swaps the second and third elements yields:
> > > >
> > > > $
> > > > f\_{\text{swap}}(\pi(S)) \neq \pi(f\_{\text{swap}}(S)).
> > > > $
> > > >
> > > > Therefore, no function in $ \mathcal{F}\_{\emptyset} $ can implement $ f\_{\text{swap}} $. This validates our t-ALN design, proving that $ \mathcal{F}\_{\text{pos}} $ possesses a strictly superior theoretical capacity for modeling the temporal structure of sleep data.

---

### Author Response · Authors · 2025-11-26

**To All Reviewers:**

We sincerely thank all reviewers for their thoughtful and constructive feedback on our submission. We genuinely appreciate the time and effort invested in helping us refine our work. We are particularly encouraged that our work is recognized as "the first to explicitly introduce an 'expert-based stage difficulty' mechanism in sleep staging tasks" (Reviewer ndSL), possessing an "intuitive design... clear and easy to follow" (Reviewer PzED), addressing a "Well-motivated problem" with "Comprehensive evaluation" and "Good visualizations" (Reviewer 1ogm), and presenting a "clean modular design" (Reviewer MXY5).

We believe S³Net makes a significant contribution to the field of sleep staging by introducing novel architectural components that explicitly model stage-specific difficulty patterns. Our framework not only achieves state-of-the-art performance but also provides interpretable insights into sleep stage transitions. We have revised the manuscript accordingly, with all changes, including the clarified contributions, highlighted in blue.

Our detailed responses address individual queries and suggestions from each reviewer. We have conducted extensive additional experiments to support our claims, covering the following key areas:

**Theoretical and Physiological Justification:**
*   **t-ALN Design:** We addressed concerns regarding t-ALN's design principles (Reviewers ndSL, PzED, 1ogm). We clarified that t-ALN serves as a critical bridge between spectral and temporal domains, functioning as an information bottleneck that ensures strict temporal alignment and forces selective attention to temporal patterns. Comprehensive visualizations and ablation studies in our manuscript validate these claims.
*   **SAE Design:** We addressed concerns about stage grouping in our SAE module (Reviewers ndSL, PzED, 1ogm). We provided theoretical explanations and strong physiological evidence justifying the partition of sleep stages into "Hard-Separated" (W, N1, N2) and "Easy-Separated" (N3, REM) groups. This grouping is further supported by statistical evidence including inter-class variance analysis, confusion entropy metrics, and comparative experiments with alternative grouping strategies. We also grounded the element-wise squaring as an energy representation in established PSG analysis principles and included experiments comparing different grouping strategies to demonstrate our design's effectiveness.

**Methodological Novelty and Effectiveness:**
We clarified distinctions between our t-ALN/SAE modules and prior works like cVAN (Reviewers PzED, ndSL, MXY5). Rigorous ablation studies compared our t-ALN against: cVA (replacing t-ALN with CVA from cVAN), Pooling-ALN (substituting dual softmax weighting with learnable pooling-based alignment), and Flat-ALN (flattening spectral maps without alignment). These experiments quantitatively demonstrate our alignment strategy's superiority. We also evaluated different sleep stage groupings within SAE and provided full ablation results across all three datasets (ISRUC-S1, ISRUC-S3, Sleep-EDF-153) to verify each component's contribution.

**Generalization and Robustness:**
To address generalization concerns (Reviewers MXY5, ndSL), we performed:
*  **Cross-dataset generalization:** Training on ISRUC-S1 and testing on ISRUC-S3, where S³Net outperformed strong baselines.
*  **Cross-domain application:** Evaluating S³Net on the WESAD dataset for single-lead PPG classification, achieving competitive performance.
*  **Missing modality experiments:** Testing robustness under single-modality settings (EEG-only, EOG-only, EMG-only), demonstrating S³Net maintains superior performance even with limited data.

**Statistic significance:**
To complete the statistical significance (Reviewer 1ogm), We conducted significance tests across in ISRUC-S3 dataset, showing that S³Net achieves statistically significant improvements over all baselines. The results, supported by both p-value and paired t-test analyses, confirm that these gains are reliable and not attributable to random variation.

**Efficiency Analysis:**
We addressed model complexity concerns (Reviewers PzED, 1ogm) with detailed comparisons of parameters, GFLOPs, training time, and inference time. Results indicate S³Net is faster and more parameter-efficient than key baselines like cVAN.

Once again, we thank all reviewers for their invaluable suggestions and insights, which have significantly contributed to the improvement of our paper. Please feel free to reach out if further clarification is required to assist in the final assessment.

---

### Author Response · Authors · 2025-11-27

# Dear ICLR 2026 AC, SAC, PC,

We are writing to express our sincere gratitude for your time and the thoughtful feedback on our S³Net submission. We are fully aware of the challenges arising from the recent unexpected situation with the ICLR 2026 review system. To support the committee and ensure the quality of the review process during this busy period, we have summarized our core contributions and detailed our responses to the reviewers' valuable points below. We hope this summary helps streamline the rebuttal process.

## Core Contributions of S³Net

To address the limitations of existing deep learning approaches in modeling subtle transitional sleep stages (N1 and N2) and efficiently bridging the temporal-spectral domain gap, we propose S³Net, a novel Stage-Aware Sleep Staging Network. Our contributions are summarized below:

- We propose t-ALN, a novel temporal alignment module that bridges the spectral-temporal representation mismatch in sleep staging through a learnable projection. Furthermore, t-ALN as an information bottleneck constrains the temporal representation, thereby implicitly preserving features relevant to sleep stage classification.
- We introduce SAE, a stage-aware experts module, which is a dynamic architecture that employs two specialized experts to address the heterogeneous complexity of sleep staging. One expert captures the subtle dynamics of transitional stages (W, N1, N2), while the other distinguishes the distinct stages of N3 and REM, leading to more robust and discriminative feature learning.
- Empirical results demonstrate that S³Net sets a new state-of-the-art on major sleep staging datasets of ISRUC-S1, ISRUC-S3, and Sleep-EDF-153. Furthermore, S³Net achieves a reduction in inference time, as experimentally validated.
- The design choices of S³Net are rigorously validated by comprehensive ablation studies, while the model's interpretability is substantiated through visualizations such as alignment maps, and analyses of expert routing behavior.

Meanwhile, building upon this contributions, we have addressed the following concerns raised by the reviewers:

## Summary of Rebuttal-Driven Revisions

In direct response to the reviewers' comments, we have made substantial revisions to the manuscript, with all changes highlighted in blue, significantly strengthening its theoretical foundation, reproducibility, and empirical validation. To facilitate your review and minimize the need to navigate through scattered discussion threads, we guarantee that the following summary includes every weakness and question raised by the reviewers, providing a comprehensive, point-by-point response to ensure no concern is left unaddressed.


### 1. Strengthened Theoretical Foundation & Justification
- **Theoretical Proof for SAE:** *To address concerns about the theoretical basis of our hierarchical design,* we derived a proof that establishes the exact conditions under which the Stage-Aware Experts (SAE) module outperforms a single classifier. This provides a rigorous mathematical guarantee for our architecture's effectiveness ($\color{brown}{\textbf{Inspired by Reviewer ndSL Q2, }}$ $\color{brown}{\textbf{ Reviewer  1ogm W1, and detail see Appendix A.17}}$).
- **Theoretical Analysis of t-ALN:** *Responding to questions about the mechanism and justification of t-ALN,* we added a detailed theoretical analysis that formally defines t-ALN as a time alignment module designed to bridge the gap between spectral and temporal representations. We theoretically verified that our strict temporal alignment is meaningful for preserving the sequential integrity of sleep stages ($\color{brown}{\textbf{Inspired by Reviewer PzED W2, Reviewer 1ogm W2, Reviewer ndSL Q2, and detail see Section 3.3, Appendix A.18}}$).
- **Physiological Basis for Expert Grouping:** *To answer questions regarding our grouping strategy (W/N1/N2 vs. N3/REM),* we incorporated transition probability matrices, support from physiological studies, and statistical evidence (such as Confusion Entropy and Inter-class Variance), firmly establishing the physiological validity of our SAE architecture ($\color{brown}{\textbf{Inspired by }}$ $\color{brown}{\textbf{ Reviewer 1ogm W1, Q1, Reviewer PzED W5, and detail see Appendix A.13}}$).

---

> ### Author Response · Authors · 2025-12-02
>
> ### 2. Enhanced Reproducibility & Clarity
> - **CST Implementation Details:** *To ensure the Cross Swin Transformer (CST) is fully reproducible and transparent,* we added a comprehensive description and complexity analysis of this core component ($\color{brown}{\textbf{Inspired by}}$ $\color{brown}{\textbf{Reviewer 1ogm}}$ $\color{brown}{\textbf{W5, and}}$ $\color{brown}{\textbf{detail see}}$ $\color{brown}{\textbf{Appendix A.3}}$).
> - **Detailed t-ALN Implementation:** *To clarify how t-ALN works and make it easier to reimplement,* we revised the details of the time-window-wise linear operation. By organizing spectral features along the time axis via a learnable projection, we demonstrate how t-ALN acts as an information bottleneck when serving as a query, implicitly preserving features critical for sleep stage classification ($\color{brown}{\textbf{Inspired by}}$ $\color{brown}{\textbf{Reviewer 1ogm}}$ $\color{brown}{\textbf{W5, Reviewer}}$ $\color{brown}{\textbf{PzED W2,}}$ $\color{brown}{\textbf{and detail}}$ $\color{brown}{\textbf{see Appendix}}$ $\color{brown}{\textbf{A.18}}$).
>
>
> ### 3. Expanded Experimental Validation
> - **Generalization to New Modalities:** *To show the model's versatility and ability to adapt to non-PSG signals,* we evaluated S³Net on the WESAD dataset using only single-lead PPG, achieving strong performance ($\color{brown}{\textbf{Inspired by}}$ $\color{brown}{\textbf{Reviewer ndSL}}$ $\color{brown}{\textbf{Q3, Reviewer}}$ $\color{brown}{\textbf{MXY5 Q2,}}$ $\color{brown}{\textbf{and detail}}$ $\color{brown}{\textbf{see Appendix}}$ $\color{brown}{\textbf{A.10}}$).
> - **Robustness Tests:** *To confirm robustness against domain shifts and sensor failures,* we conducted rigorous cross-dataset evaluations (training on ISRUC-S1, testing on ISRUC-S3) and missing-modality tests (EEG-only, EOG-only, EMG-only). These experiments confirm that S³Net achieves superior performance in both single-modality and cross-dataset scenarios ($\color{brown}{\textbf{Inspired by Reviewer MXY5 Q2, and}}$ $\color{brown}{\textbf{detail see}}$ $\color{brown}{\textbf{Appendix A.11,}}$ $\color{brown}{\textbf{A.12}}$).
> - **Alignment Strategies:** *To address concerns about the comparison with other fusion mechanisms (e.g., cVAN) and the justification for t-ALN,* we performed controlled comparisons with other alignment modules (e.g., CVA, Pooling-ALN, Flat-ALN). The results demonstrate the superior effectiveness of t-ALN in bridging representation mismatches ($\color{brown}{\textbf{Inspired by}}$ $\color{brown}{\textbf{Reviewer 1ogm}}$ $\color{brown}{\textbf{W2, Q2,}}$ $\color{brown}{\textbf{Reviewer PzED}}$ $\color{brown}{\textbf{W1, W3,}}$ $\color{brown}{\textbf{Q1, Reviewer}}$ $\color{brown}{\textbf{MXY5 Q1}}$).
> - **Computational Efficiency:** *To address concerns about computational costs,* we provided a full complexity analysis (Parameters, GFLOPs, training/inference times), confirming that S³Net achieves state-of-the-art performance without compromising efficiency ($\color{brown}{\textbf{Inspired by}}$ $\color{brown}{\textbf{Reviewer ndSL}}$ $\color{brown}{\textbf{Q2, Reviewer}}$ $\color{brown}{\textbf{PzED W4,}}$ $\color{brown}{\textbf{Q3, Reviewer}}$ $\color{brown}{\textbf{1ogm W3,}}$ $\color{brown}{\textbf{Q4, and}}$ $\color{brown}{\textbf{detail see}}$ $\color{brown}{\textbf{Appendix A.6}}$).
> - **Statistical Significance:** *To validate the reliability of our performance improvements,* we performed paired t-tests, confirming that the gains achieved by S³Net are statistically significant (p < 0.05) ($\color{brown}{\textbf{Inspired by}}$ $\color{brown}{\textbf{Reviewer 1ogm}}$ $\color{brown}{\textbf{W3, Q5,}}$ $\color{brown}{\textbf{Reviewer PzED}}$ $\color{brown}{\textbf{W4}}$).
> - **Ablation study on other dataset:** *Responding to requests for broader ablation studies,* we extended our ablation analysis to Sleep-EDF-153 and ISRUC-S1, confirming that each module of S³Net is indispensable across different datasets ($\color{brown}{\textbf{Inspired by}}$ $\color{brown}{\textbf{Reviewer 1ogm}}$ $\color{brown}{\textbf{W3, and}}$ $\color{brown}{\textbf{detail see}}$ $\color{brown}{\textbf{Section 4.5}}$).
> - **Population Stability:** *To verify generalizability across different groups,* we analyzed transition matrices across different age groups, confirming the stability and applicability of our stage-aware grouping strategy ($\color{brown}{\textbf{Inspired by}}$ $\color{brown}{\textbf{Reviewer 1ogm}}$ $\color{brown}{\textbf{Q3, and}}$ $\color{brown}{\textbf{detail see}}$ $\color{brown}{\textbf{Appendix A.16}}$).
> - **Training Dynamics:** *To demonstrate convergence stability,* we provided detailed visualizations of training/validation loss and accuracy curves ($\color{brown}{\textbf{Inspired by}}$ $\color{brown}{\textbf{Reviewer 1ogm}}$ $\color{brown}{\textbf{W5, and}}$ $\color{brown}{\textbf{detail see}}$ $\color{brown}{\textbf{Appendix A.14,}}$ $\color{brown}{\textbf{A.15}}$).

---

> ### Author Response · Authors · 2025-12-03
>
> ### 4. Clarifications on Methodological Misunderstandings
> - **Methodological Novelty:** *To address questions about novelty compared to generic MoE or previous alignment approaches,* we clarified that S³Net’s contribution lies in the principled integration of physiological priors via SAE and the specific query-bottleneck design of t-ALN ($\color{brown}{\textbf{Inspired by}}$ $\color{brown}{\textbf{Reviewer PzED}}$ $\color{brown}{\textbf{W1,W3, Reviewer}}$ $\color{brown}{\textbf{1ogm W4,}}$ $\color{brown}{\textbf{and detail}}$ $\color{brown}{\textbf{see Section}}$ $\color{brown}{\textbf{3.3, 3.4}}$).
> - **Energy Representation:** *To justify our signal processing choices,* we clarified that element-wise squaring is a standard practice for energy estimation, supported by theoretical signal analysis ($\color{brown}{\textbf{Inspired by}}$ $\color{brown}{\textbf{Reviewer 1ogm}}$ $\color{brown}{\textbf{W2, and}}$ $\color{brown}{\textbf{detail see}}$ $\color{brown}{\textbf{Section 3.3}}$).
> - **Fair Baseline Comparison:** *Correcting a misunderstanding regarding experimental fairness,* we confirmed that S³Net uses the standard channel setup (10 for ISRUC, 2 for Sleep-EDF) matching all baselines ($\color{brown}{\textbf{Inspired by}}$ $\color{brown}{\textbf{Reviewer 1ogm}}$ $\color{brown}{\textbf{W3, and}}$ $\color{brown}{\textbf{detail see}}$ $\color{brown}{\textbf{Section 4.2}}$).
> - **Visualization & Interpretability:** *Responding to requests for qualitative evidence,* we highlighted existing visualizations (alignment maps, t-SNE) and ablation studies that qualitatively demonstrate the efficacy of t-ALN and SAE ($\color{brown}{\textbf{Inspired by}}$ $\color{brown}{\textbf{Reviewer PzED}}$ $\color{brown}{\textbf{Q2, and}}$ $\color{brown}{\textbf{detail see}}$ $\color{brown}{\textbf{Section 4.5}}$).
> - **Hyperparameter Analysis:** *To address concerns about auxiliary loss weights and expert count,* we clarified that the two-expert configuration is empirically optimal (Figure 8) and analyzed gradient dynamics to confirm $\alpha=1$ as the optimal balance ($\color{brown}{\textbf{Inspired by}}$ $\color{brown}{\textbf{Reviewer 1ogm}}$ $\color{brown}{\textbf{W1, W5,}}$ $\color{brown}{\textbf{Reviewer MXY5}}$ $\color{brown}{\textbf{Q1, and}}$ $\color{brown}{\textbf{detail see}}$ $\color{brown}{\textbf{Section 4.5}}$).
>
> During the rebuttal phase, we focused on thoroughly addressing the reviewers' constructive feedback by deriving additional theoretical proofs, conducting extensive supplementary experiments, and refining our manuscript. We are grateful that these revisions have been positively received, noting that $\color{brown}{\textbf{Reviewer PzED}}$ $\color{brown}{\textbf{has acknowledged}}$ $\color{brown}{\textbf{our response}}$ $\color{brown}{\textbf{and explicitly}}$ $\color{brown}{\textbf{promised to}}$ $\color{brown}{\textbf{raise the}}$ $\color{brown}{\textbf{score}}$. We sincerely thank the AC and all reviewers for their time and valuable suggestions, which have been instrumental in improving the quality of our work.
>
> Best regards,
>
> The Authors

---

### Note · Program_Chairs · 2026-01-17
**Submission Desk Rejected by Program Chairs**

The following references in this submission do not refer to real documents and/or have major errors in bibliographic information:

 R. Yao et al. Drfuse: Shared-and-specific representation fusion for multimodal sleep staging. arXiv / conference (2024), 2024. URL https://arxiv.org/abs/2403.06197.